# ALIGNING LATENT SPACES WITH FLOW PRIORS

## ABSTRACT

This paper presents a novel framework for aligning learnable latent spaces to arbitrary prior distributions by leveraging flow-matching generative models as priors. Our method first pretrains a flow-matching model on the prior features to capture the underlying distribution. This fixed flow model subsequently regularizes the latent space via an alignment loss, which reformulates the flow matching objective to treat the latents as optimization targets. We formally prove that minimizing this alignment loss establishes a computationally tractable surrogate objective for maximizing a variational lower bound on the log-likelihood of latents under the prior distribution. Notably, the proposed method eliminates expensive likelihood evaluations and avoids ODE solving during optimization. As a proof of concept, we demonstrate in a controlled setting that the alignment loss landscape closely approximates the negative log-likelihood of the prior. We further validate the effectiveness of our approach by regularizing the latent spaces of autoencoders in large-scale ImageNet image generation, with diverse prior distributions, accompanied by detailed discussions and ablation studies. With both theoretical and empirical validation, our framework paves a new way for latent space alignment.

## 1 INTRODUCTION

Latent models like autoencoders (AEs) are a cornerstone of modern machine learning (Hinton & Salakhutdinov, 2006; Baldi, 2012; Li et al., 2023; Chen & Guo, 2023; Mienye & Swart, 2025). These models typically map high-dimensional observations to a lower-dimensional latent space, aiming to capture salient features and dependencies (Liou et al., 2014; Meng et al., 2017). A highly desirable property of latent models is that the latent space should have structural properties, such as being close to a predefined prior distribution (Salah et al., 2011; Kingma & Welling, 2014; Yao et al., 2025; Chen et al., 2024). Such structure can incorporate domain-specific prior knowledge (Khemakhem et al., 2020; Raissi et al., 2019), enhance the interpretability of the latent space (Higgins et al., 2017; Chen et al., 2016; Kim & Mnih, 2018), and facilitate latent space generation (Rombach et al., 2022; Li et al., 2024a; Leng et al., 2025; Wen et al., 2025; Yu et al., 2025). For example, in latent generative models, encouraging latents of the autoencoder to follow a structured prior enables high-quality synthesis.

Traditional approaches to enforcing distributional conformity often involve minimizing divergences like the Kullback-Leibler (KL) divergence (Kingma & Welling, 2014; Rombach et al., 2022). However, KL can be restrictive, particularly when the prior is only implicitly defined (e.g., by samples). In latent generative modeling, the latent space is usually regularized with known prior distributions, such as the Gaussian distribution for Variational Autoencoders (VAE) (Kingma & Welling, 2014; Esser et al., 2021), and the categorical distribution for Vector Quantized VAE (VQ-VAE) (Van Den Oord et al., 2017). Recent works (Qiu et al., 2025; Li et al., 2024b;c; Chen et al., 2025a; Yao et al., 2025; Chen et al., 2024) have proposed to use pre-trained feature extractors as samples that define the prior over latents and directly optimize the latent distances, which are shown to be effective but computationally expensive and require per-sample features.

Recent advances in flow-matching (FM) generative models (Lipman et al., 2023; Liu et al., 2023) offer a promising avenue to capture complex prior distributions. In this work, we address the question: *Can we efficiently align a learnable latent space to an arbitrary prior distribution using a pre-trained FM model as a prior?* We answer this question affirmatively by proposing a novel framework that leverages a pre-trained FM model to define a computationally tractable alignment loss, which effectively guides the latents towards the prior distribution.

Figure 1: (a) Conventional alignment works with only known priors (e.g., Gaussian or categorical) using KL or cross-entropy loss. (b) Our proposed method can align the latent distribution to an **arbitrary** prior distribution captured by a pre-trained flow model.

Our proposed approach unfolds in a two-stage process as illustrated in Fig.1. The first stage involves pretraining an FM model on the desired prior features, allowing it to learn the mapping from a base distribution (e.g., Gaussian) to the prior distribution. Once this flow model accurately captures the prior distribution, its parameters are fixed. In the second stage, this flow model serves as a prior to regularize a learnable latent space, for instance, the output of the encoder in an AE. This regularization is achieved by minimizing an alignment loss, which ingeniously adapts the standard FM objective by treating the learnable latents as the target. This pipeline provides an efficient mechanism to guide the latent space towards the desired prior structure without requiring direct comparison to prior samples or expensive likelihood evaluations of the flow model.

We theoretically justify our method by connecting the alignment loss to the maximum likelihood estimation of the latents under the prior distribution. While directly maximizing this likelihood under a flow model is often computationally prohibitive due to the need to evaluate the trace of Jacobian determinants and solve an ordinary differential equation (ODE) for each step, our alignment loss offers a more tractable alternative. We formally demonstrate that minimizing this loss serves as a computationally efficient proxy for maximizing a variational lower bound on the log-likelihood of the latents under the flow-defined prior distribution.

Our framework offers three key advantages. First, our approach enables alignment to **arbitrary prior distributions**, even those implicitly defined by samples, overcoming the limitations of conventional methods that require explicit parametric priors. Second, the alignment loss acts as a **direct surrogate** for the log-likelihood of the latents under the prior distribution, providing a theoretically grounded objective that avoids heuristic metrics like cosine similarity used in per-sample feature matching (Chen et al., 2025a; Yao et al., 2025; Chen et al., 2024). Third, our framework is **computationally lightweight**, requiring only a single forward pass through the pre-trained flow model during training, thereby bypassing the need for expensive adversarial optimization (Goodfellow et al., 2020), likelihood evaluations, or per-sample feature extraction (Qiu et al., 2025; Li et al., 2024b;c).

We empirically validate the efficacy of our proposed alignment strategy through a series of experiments. We start with illustrative experiments in a controlled toy setting using a mixture of Gaussians to confirm that our alignment loss landscape indeed serves as a proxy for the log-likelihood of the latents under the prior distribution. Then we demonstrate the scalability of our approach by conducting large-scale latent image generation experiments on ImageNet (Deng et al., 2009) with diverse prior distributions. Detailed discussions and ablation studies are provided to underscore the robustness and effectiveness.

We believe this method offers a powerful and flexible tool for incorporating rich distributional priors into latent models. Our work paves the way for more flexible and powerful structured representation learning, and we anticipate its application and extension in various domains requiring distributional structure control over latent spaces.

## 2 RELATED WORK

### 2.1 FLOW-BASED MODELS

Flow-based generative models have emerged as a powerful class of generative models (Esser et al., 2024; Labs, 2024; Chen et al., 2025b; Kingma & Dhariwal, 2018; Zhai et al., 2025; Zhao et al., 2024; Shin et al., 2025). They first appeared as discrete normalizing flows such as NICE and Real NVP,

which introduced additive and affine coupling layers for exact invertibility, with Glow scaling these designs (Dinh et al., 2014; 2017; Kingma & Dhariwal, 2018). This line of work later branched into continuous normalizing flows (CNFs) that learn an ODE-driven invertible mapping between a simple base distribution and complex data (Chen et al., 2018; Grathwohl et al., 2019). A notable recent development is Flow Matching (FM) (Lipman et al., 2023; Albergo & Vanden-Eijnden, 2023; Liu et al., 2023; Neklyudov et al., 2023; Heitz et al., 2023; Tong et al., 2023), which simplifies training by regressing a vector field against a target field from sample pairs, and it clarifies links between flows and diffusion models via probability-flow formulations. In ICTM (Zhang et al., 2024), flow priors of generative models have been employed for MAP estimation to solve linear inverse problems. Our work leverages flow-based models to learn complex distributions as a prior for latent space alignment.

## 2.2 Latent Space Alignment

The alignment of latent spaces with predefined distributions is a crucial aspect of representation learning. In VAE (Kingma & Welling, 2014), the latent space is typically regularized to follow a standard Gaussian distribution. Several approaches have been proposed to use more flexible priors, such as hierarchical VAEs (Sønderby et al., 2016; Vahdat & Kautz, 2020) or VAEs with inverse autoregressive flow (IAF) priors (Kingma et al., 2016). Another line of work focuses on aligning latent spaces with features extracted from pre-trained models (Qiu et al., 2025; Li et al., 2024b;c; Chen et al., 2025a; Yao et al., 2025; Chen et al., 2024; Kim et al., 2025; Zha et al., 2024). Our method differs by utilizing a pre-trained flow model to define an expressive target and a novel alignment loss, avoiding expensive likelihoods evaluation, adversarial training, or direct per-sample feature comparison.

## 3 Preliminaries

### 3.1 Flow Matching

We consider an ODE defined by a time-dependent velocity field $\boldsymbol{u}(\boldsymbol{x}_t, t)$:

$$\frac{\mathrm{d}\boldsymbol{x}_t}{\mathrm{d}t} = \boldsymbol{u}(\boldsymbol{x}_t, t), \quad \boldsymbol{x}_0 \sim p_{\text{init}}, \quad \boldsymbol{x}_1 \sim p_{\text{prior}} \tag{1}$$

where $p_{\text{init}}$ is a simple base (e.g., $\mathcal{N}(\boldsymbol{0}, \boldsymbol{I})$) and $p_{\text{prior}}$ is the target distribution. The state $\boldsymbol{x}_t \in \mathbb{R}^d$ evolves from $\boldsymbol{x}_0$ to $\boldsymbol{x}_1$ as $t$ goes from 0 to 1, assuming $\boldsymbol{u}$ is Lipschitz in $\boldsymbol{x}$ and continuous in $t$.

Since $\boldsymbol{u}$ is unknown, we approximate it with a neural network $\boldsymbol{v}_\theta(\boldsymbol{x}_t, t)$:

$$\frac{\mathrm{d}\boldsymbol{x}_t}{\mathrm{d}t} = \boldsymbol{v}_\theta(\boldsymbol{x}_t, t), \quad \boldsymbol{x}_0 \sim p_{\text{init}} \tag{2}$$

The solution $\boldsymbol{x}_t = \boldsymbol{\Phi}_t^\theta(\boldsymbol{x}_0)$ defines a flow from $\boldsymbol{x}_0$, aiming to match $\boldsymbol{x}_1 = \boldsymbol{\Phi}_1^\theta(\boldsymbol{x}_0)$ with $p_{\text{prior}}$.

Flow matching (Lipman et al., 2023; Liu et al., 2023) trains $\boldsymbol{v}_\theta$ to approximate a target velocity field. This is built from a probability path $p_t(\boldsymbol{x})$ interpolating between $p_{\text{init}}$ and $p_{\text{prior}}$. A common choice is a conditional path $\boldsymbol{x}_t(\boldsymbol{x}_0, \boldsymbol{x}_1)$ with linear interpolation: $\boldsymbol{x}_t = (1 - t)\boldsymbol{x}_0 + t\boldsymbol{x}_1$. The associated velocity is $\boldsymbol{u}_t(\boldsymbol{x}_t | \boldsymbol{x}_0, \boldsymbol{x}_1) = \boldsymbol{x}_1 - \boldsymbol{x}_0$. The flow matching loss is:

$$\mathcal{L}_{\text{FM}}(\theta) = \mathbb{E}_{t, \boldsymbol{x}_0, \boldsymbol{x}_1} \left[ \|\boldsymbol{v}_\theta((1 - t)\boldsymbol{x}_0 + t\boldsymbol{x}_1, t) - (\boldsymbol{x}_1 - \boldsymbol{x}_0)\|^2 \right] \tag{3}$$

In this paper, we assume $\boldsymbol{v}_\theta$ is pre-trained, fixed, and optimal, i.e., it exactly minimizes Eq. (3) so that $\boldsymbol{v}_\theta((1 - t)\boldsymbol{x}_0 + t\boldsymbol{x}_1, t) = \boldsymbol{x}_1 - \boldsymbol{x}_0$ for all $(\boldsymbol{x}_0, \boldsymbol{x}_1, t)$. Such a $\boldsymbol{v}_\theta$ can serve as a regularizer to align latents with the prior distribution $p_{\text{prior}}$.

### 3.2 Likelihood Estimation with Flow Priors

Let $p_1^{\boldsymbol{v}_\theta}(\boldsymbol{x}_1)$ denote the probability density at $t = 1$ induced by the flow model $\boldsymbol{v}_\theta$ evolving from $p_{\text{init}}$. Using the instantaneous change of variables formula, the log-likelihood of a sample $\boldsymbol{x}_1$ under this model can be computed by (Chen et al., 2018; Grathwohl et al., 2019):

$$\log p_1^{\boldsymbol{v}_\theta}(\boldsymbol{x}_1) = \log p_{\text{init}}(\boldsymbol{x}_0) - \int_0^1 \text{Tr}(\nabla_{\boldsymbol{x}} \boldsymbol{v}_\theta(\boldsymbol{x}_s, s)) \mathrm{d}s \tag{4}$$

Here, $\boldsymbol{x}_s = \boldsymbol{\Phi}_s^\theta(\boldsymbol{x}_0)$ is the trajectory generated by $\boldsymbol{v}_\theta$ starting from $\boldsymbol{x}_0$ and ending at $\boldsymbol{x}_1 = \boldsymbol{\Phi}_1^\theta(\boldsymbol{x}_0)$. Thus, $\boldsymbol{x}_0 = (\boldsymbol{\Phi}_1^\theta)^{-1}(\boldsymbol{x}_1)$ is obtained by flowing $\boldsymbol{x}_1$ backward in time to $t = 0$. Given a pre-trained flow model $\boldsymbol{v}_\theta$ that maps $p_{\text{init}}$ (e.g., Gaussian noise) to a prior distribution (e.g., prior features), one can align new input samples with these prior features by maximizing their log-likelihood under $p_1^{\boldsymbol{v}_\theta}$. However, computing Eq. (4) is often computationally expensive, primarily due to the trace of the Jacobian term $(\text{Tr}(\nabla_{\boldsymbol{x}} \boldsymbol{v}_\theta))$ and the need for an ODE solver. In this paper, we demonstrate that a similar alignment objective can be achieved by minimizing the flow matching loss Eq. (3) with respect to $\boldsymbol{x}_1$, treating $\boldsymbol{x}_1$ as a variable to be optimized rather than a fixed sample from $p_{\text{prior}}$.

## 4 METHOD

In this paper, we aim to align a learnable latent space, whose latents are denoted by $\boldsymbol{z}$, to a prior distribution $p_{\text{prior}}$. We first describe the overall pipeline in Sec. 4.1. Our method leverages a pre-trained FM model to implicitly capture $p_{\text{prior}}$ and subsequently align the latents $\boldsymbol{z}$ with it. Then, we provide an intuitive explanation in Sec. 4.2 and a formal proof of the proposed method in Sec. 4.3.

### 4.1 PIPELINE

In an autoencoder (AE) with encoder $E_\phi$ and decoder $D_\psi$, we want the latent codes of the encoder to conform to the distribution of features from a pre-trained feature extractor. More generally, let $\boldsymbol{z} \in \mathbb{R}^{d_1}$ denote a sample from a learnable latent space, produced by a parametric model $E_\phi$. Let $\boldsymbol{x} \in \mathbb{R}^{d_2}$ be a sample from the prior feature space, characterized by an underlying distribution $\boldsymbol{x} \sim p_{\text{prior}}(\boldsymbol{x})$. Our objective is to train $E_\phi$ such that the distribution of its outputs, $p_\phi(\boldsymbol{z})$, aligns with $p_{\text{prior}}(\boldsymbol{x})$. This alignment can be formulated as maximizing the likelihood of $\boldsymbol{z}$ under $p_{\text{prior}}$.

**Addressing the Dimension Mismatch** A challenge arises if the latent space dimension $d_1$ differs from the prior feature space dimension $d_2$. To address this, we employ fixed (non-learnable) linear projections to map prior features $\boldsymbol{x}$ from $\mathbb{R}^{d_2}$ to $\mathbb{R}^{d_1}$. For simplicity, we continue to denote the projected features and their distribution as $\boldsymbol{x}$ and $p_{\text{prior}}$, respectively. We consider three alternative projection operators: *Random Projection*, *Average Pooling*, and *PCA*. We ablate these methods in Sec. 5.3 and select random projection as the default due to its simplicity and empirical effectiveness.

The use of linear projection is theoretically supported by the Johnson-Lindenstrauss (JL) lemma (Johnson et al., 1984). The JL lemma states that for a set of $N$ points in $\mathbb{R}^{d_2}$, a random linear mapping can preserve all pairwise Euclidean distances within a multiplicative distortion factor. The linear projection is defined by a matrix $\boldsymbol{W} \in \mathbb{R}^{d_1 \times d_2}$. Assuming the prior features $\boldsymbol{x}$ are appropriately normalized, we initialize the projection matrix $\boldsymbol{W}$ by sampling its entries from $\mathcal{N}(0, 1/d_2)$. This scaling helps ensure that the components of $\boldsymbol{W}\boldsymbol{x}$ have approximately unit variance if the components of $\boldsymbol{x}$ are uncorrelated, thereby preserving key statistical properties.

**Flow Prior Estimation** With the projected prior features $\boldsymbol{x} \sim p_{\text{prior}}$, we first train an FM model $\boldsymbol{v}_\theta : \mathbb{R}^{d_1} \times [0, 1] \to \mathbb{R}^{d_1}$, parameterized by $\theta$. This model is trained using the flow matching objective Eq. (3), where the dimension $d$ is set to $d_1$, $\boldsymbol{x}_0 \sim \mathcal{N}(\boldsymbol{0}, \boldsymbol{I})$, and $\boldsymbol{x}_1$ is replaced by samples $\boldsymbol{x}$ from $p_{\text{prior}}$. After training, the parameters $\theta$ of the FM model $\boldsymbol{v}_\theta$ are frozen. This fixed $\boldsymbol{v}_\theta$ implicitly defines a generative process capable of transforming samples from $p_{\text{init}}$ (now in $\mathbb{R}^{d_1}$) into samples that approximate $p_{\text{prior}}$. It captures the underlying prior distribution and serves as a distributional prior for aligning the latent space.

**Latent Space Regularization** Once $\boldsymbol{v}_\theta$ is trained and its parameters fixed, we use it to regularize the learnable latents $\boldsymbol{z}$. The goal is to encourage the distribution $p_\phi(\boldsymbol{z})$ to conform to $p_{\text{prior}}$ as captured by $\boldsymbol{v}_\theta$. For each $\boldsymbol{z}$ produced by $E_\phi$, we incorporate the flow matching objective described in Eq. (3) into the training objective of $E_\phi$:

$$\mathcal{L}_{\text{align}}(\boldsymbol{z}; \theta) = \mathop{\mathbb{E}}_{t \sim \mathcal{U}[0,1], \boldsymbol{x}_0 \sim p_{\text{init}}(\boldsymbol{x}_0)} \left[ \| \boldsymbol{v}_\theta((1-t)\boldsymbol{x}_0 + t\boldsymbol{z}, t) - (\boldsymbol{z} - \boldsymbol{x}_0) \|^2 \right] \quad (5)$$

Here, $p_{\text{init}}$ is the same $d_1$-dimensional base distribution $\mathcal{N}(\boldsymbol{0}, \boldsymbol{I})$ used for training $\boldsymbol{v}_\theta$. In Sec. 4.3, we formally prove that minimizing Eq. (5) with respect to $\boldsymbol{z}$ serves as a proxy to maximizing a lower bound on the log-likelihood $\log p_1^{\boldsymbol{v}_\theta}(\boldsymbol{z})$. This establishes that minimizing $\mathcal{L}_{\text{align}}(\boldsymbol{z}; \theta)$ effectively trains $E_\phi$ such that its outputs $\boldsymbol{z}$ align with the distribution of the prior features $\boldsymbol{x}$.

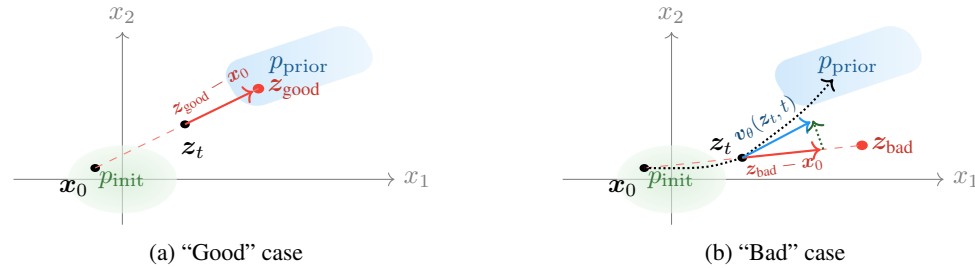

(a) "Good" case
(b) "Bad" case

Figure 2: Intuitive illustration of latent space alignment via flow matching, best viewed in color. (a) A "good" $z_{\mathrm{good}}$ in $p_{\mathrm{prior}}$ (blue) aligns the straight path velocity (red solid arrow) with the pre-trained flow model's velocity $v_\theta(z_t, t)$ (overlapped and omitted), yielding low loss. (b) A "bad" $z_{\mathrm{bad}}$ outside $p_{\mathrm{prior}}$ causes a mismatch between the path velocity and $v_\theta(z_t, t)$ (blue solid arrow), resulting in high loss. Minimizing this loss steers $z_{\mathrm{bad}}$ to $p_{\mathrm{prior}}$ (green dotted arrow).

The key insight is that the pre-trained velocity field $v_\theta$ encapsulates the dynamics that transport probability mass from the base distribution $p_{\mathrm{init}}$ to the prior distribution $p_{\mathrm{prior}}$ along linear paths. By minimizing $\mathcal{L}_{\mathrm{align}}(z; \theta)$, we penalize latents $z$ that do not conform to these learned dynamics—that is, $z$ values for which the path $(1 - t)x_0 + tz$ is not "natural" under $v_\theta$. This procedure shapes $p_\phi(z)$ to match $p_{\mathrm{prior}}$ without requiring explicit computation of potentially intractable likelihoods, relying instead on the computationally efficient flow matching objective.

## 4.2 INTUITIVE EXPLANATION

Our alignment method leverages the pre-trained flow model, $v_\theta$, as an expert on the prior feature distribution $p_{\mathrm{prior}}$. Having been well trained, $v_\theta$ precisely captures the dynamics required to transform initial noise samples $x_0$ into prior features $x$ along straight interpolation paths. Specifically, it has learned to predict the exact velocity $x - x_0$ at any point $(1 - t)x_0 + tx$ along such a path. This effectively means $v_\theta$ can validate whether a given trajectory from noise is characteristic of those leading to the true prior distribution.

We utilize this knowledge to shape the distribution of our learnable latents $z$. The alignment loss, $\mathcal{L}_{\mathrm{align}}(z; \theta)$, challenges $v_\theta$: for a given $z$ and a random $x_0$, it asks whether the velocity field predicted by $v_\theta$ along the straight path $(1 - t)x_0 + tz$ matches the path's inherent velocity, $z - x_0$. If $z$ is statistically similar to samples from $p_{\mathrm{prior}}$, this match will be close, resulting in a low loss. Conversely, a significant mismatch indicates that $z$ is not a plausible prior according to the learned dynamics, yielding a high loss. By minimizing this loss (by optimizing the generator $E_\phi$ that produces $z$), we iteratively guide $z$ towards regions where its connecting path from noise is endorsed by $v_\theta$. As depicted in Fig. 2, this process progressively aligns the distribution of $z$ (red) with the prior distribution $p_{\mathrm{prior}}$ (blue), achieving distributional conformity.

## 4.3 RELATING THE ALIGNMENT LOSS TO AN ELBO ON LOG-LIKELIHOOD

In this section, we demonstrate that minimizing the alignment loss $\mathcal{L}_{\mathrm{align}}(z; \theta)$ (Eq. (5)) with respect to a given $z \in \mathbb{R}^{d_1}$ corresponds to maximizing a variational lower bound (ELBO) on the log-likelihood $\log p_1^{v_\theta}(z)$. Here, $p_1^{v_\theta}(z)$ denotes the probability density at $t = 1$ induced by the ODE dynamics $\frac{dz_t}{dt} = v_\theta(z_t, t)$, with $z_0 \sim p_{\mathrm{init}}$.

**Proposition 1.** *Let $v_\theta : \mathbb{R}^{d_1} \times [0, 1] \to \mathbb{R}^{d_1}$ be a given velocity field, and $p_{\mathrm{init}}$ be a base distribution. For $z \in \mathbb{R}^{d_1}$, the log-likelihood $\log p_1^{v_\theta}(z)$ is lower-bounded as:*

$$\log p_1^{v_\theta}(z) \geq C(z) - \lambda \mathcal{L}_{align}(z; \theta), \tag{6}$$

*where $\lambda > 0$ is a constant, $\mathcal{L}_{align}(z; \theta)$ is defined in Eq. (5), and $C(z)$ is dependent on $z$ and $v_\theta$.*

*Proof.* We establish this result by constructing a specific variational lower bound on $\log p_1^{v_\theta}(z)$. Variational lower bounds for log-likelihoods in continuous-time generative models can be constructed by introducing a proposal distribution for the latents that could generate $z$. Consider a family of

"proposal" paths (Lipman et al., 2023), which are straight lines interpolating from an initial point $\boldsymbol{x}_0 \sim p_{\text{init}}$ to the given point $\boldsymbol{z}$:

$$\boldsymbol{z}_s(\boldsymbol{x}_0, \boldsymbol{z}) = (1-s)\boldsymbol{x}_0 + s\boldsymbol{z}, \quad s \in [0,1] \tag{7}$$

The velocity of such a path is constant: $\dot{\boldsymbol{z}}_s(\boldsymbol{x}_0, \boldsymbol{z}) = \boldsymbol{z} - \boldsymbol{x}_0$. We adopt a variational distribution over the initial states $\boldsymbol{x}_0$, conditioned on $\boldsymbol{z}$, as $q(\boldsymbol{x}_0|\boldsymbol{z}) = p_{\text{init}}(\boldsymbol{x}_0)$. That is, we consider initial states drawn from the base, irrespective of $\boldsymbol{z}$ for the functional form of $q$.

A known variational lower bound on $\log p_1^{\boldsymbol{v}_\theta}(\boldsymbol{z})$ (Chen et al., 2018; Grathwohl et al., 2019; Liu et al., 2023) can be written as:

$$\log p_1^{\boldsymbol{v}_\theta}(\boldsymbol{z}) \geq \mathbb{E}_{\boldsymbol{x}_0 \sim q(\cdot|\boldsymbol{z})} \left[ \log p_{\text{init}}(\boldsymbol{x}_0) - \int_0^1 \left( \lambda_s \|\dot{\boldsymbol{z}}_s(\boldsymbol{x}_0, \boldsymbol{z}) - \boldsymbol{v}_\theta(\boldsymbol{z}_s(\boldsymbol{x}_0, \boldsymbol{z}), s)\|^2 \right) \mathrm{d}s \right.$$

$$\left. - \log q(\boldsymbol{x}_0|\boldsymbol{z}) - \int_0^1 \left( \mathrm{Tr}(\nabla_{\boldsymbol{z}_s} \boldsymbol{v}_\theta(\boldsymbol{z}_s(\boldsymbol{x}_0, \boldsymbol{z}), s)) \right) \mathrm{d}s \right] \tag{8}$$

Here, $\lambda_s > 0$ is a time-dependent weighting factor. For simplicity and consistency with the definition of $\mathcal{L}_{\text{align}}$ (Eq. (5)), we set $\lambda_s = \lambda = 1$ for all $s \in [0,1]$. With $q(\boldsymbol{x}_0|\boldsymbol{z}) = p_{\text{init}}(\boldsymbol{x}_0)$, the term $\log p_{\text{init}}(\boldsymbol{x}_0) - \log q(\boldsymbol{x}_0|\boldsymbol{z})$ vanishes. Substituting the expressions for $\boldsymbol{z}_s(\boldsymbol{x}_0, \boldsymbol{z})$ from Eq. (7) and its velocity $\dot{\boldsymbol{z}}_s(\boldsymbol{x}_0, \boldsymbol{z}) = \boldsymbol{z} - \boldsymbol{x}_0$:

$$\log p_1^{\boldsymbol{v}_\theta}(\boldsymbol{z}) \geq -\mathbb{E}_{\boldsymbol{x}_0 \sim p_{\text{init}}} \left[ \int_0^1 \mathrm{Tr}(\nabla_{\boldsymbol{z}} \boldsymbol{v}_\theta((1-s)\boldsymbol{x}_0 + s\boldsymbol{z}, s)) \mathrm{d}s \right]$$

$$- \mathbb{E}_{\boldsymbol{x}_0 \sim p_{\text{init}}} \left[ \int_0^1 \|(\boldsymbol{z} - \boldsymbol{x}_0) - \boldsymbol{v}_\theta((1-s)\boldsymbol{x}_0 + s\boldsymbol{z}, s)\|^2 \mathrm{d}s \right] \tag{9}$$

The second term in this inequality matches the definition of $\mathcal{L}_{\text{align}}(\boldsymbol{z}; \theta)$ (Eq. (5)). Let us define the first term of the ELBO's right-hand side. To maintain consistency with the expectation over time in $\mathcal{L}_{\text{align}}$, we can write:

$$C(\boldsymbol{z}) = -\mathbb{E}_{s \sim \mathcal{U}[0,1], \boldsymbol{x}_0 \sim p_{\text{init}}} \left[ \mathrm{Tr}(\nabla_{\boldsymbol{z}} \boldsymbol{v}_\theta((1-s)\boldsymbol{x}_0 + s\boldsymbol{z}, s)) \right] \tag{10}$$

So, the ELBO (Eq. (9)) can be expressed as:

$$\log p_1^{\boldsymbol{v}_\theta}(\boldsymbol{z}) \geq C(\boldsymbol{z}) - \mathcal{L}_{\text{align}}(\boldsymbol{z}; \theta) \tag{11}$$

This concludes the derivation of the lower bound as stated in the proposition (with $\lambda = 1$). $\qquad\square$

**Interpretation and Significance** The inequality (11) demonstrates that maximizing the derived lower bound with respect to $\boldsymbol{z}$ involves two parts: maximizing $C(\boldsymbol{z})$ and minimizing $\mathcal{L}_{\text{align}}(\boldsymbol{z}; \theta)$. The term $\mathcal{L}_{\text{align}}(\boldsymbol{z}; \theta)$ directly measures how well the velocity field $\boldsymbol{v}_\theta$ predicts the velocity of that straight path, i.e., $\boldsymbol{z} - \boldsymbol{x}_0$. Minimizing this term forces $\boldsymbol{z}$ into regions where it behaves like a point reachable from $p_{\text{init}}$ via a straight path whose dynamics are consistent with the learned $\boldsymbol{v}_\theta$. This is precisely the behavior expected if $\boldsymbol{z}$ were a sample from the distribution $p_{\text{prior}}$.

The term $C(\boldsymbol{z})$ represents the expected negative trace of the Jacobian of $\boldsymbol{v}_\theta$, averaged over the chosen straight variational paths. By minimizing $\mathcal{L}_{\text{align}}(\boldsymbol{z}; \theta)$, we are not strictly maximizing the ELBO in Eq. (11) with respect to $\boldsymbol{z}$. Instead, we are optimizing a crucial component of it that directly enforces consistency with the learned flow dynamics. We analyze the behavior of $C(\boldsymbol{z})$ in Appendix A to show that if $\boldsymbol{z}$ aligns with a more concentrated prior distribution (making $\mathcal{L}_{\text{align}}(\boldsymbol{z}; \theta)$ small), $C(\boldsymbol{z})$ tends to be positive and larger, contributing favorably to the ELBO.

**Assumption 1** (Optimality of $\boldsymbol{v}_\theta$). *The velocity field $\boldsymbol{v}_\theta : \mathbb{R}^{d_1} \times [0,1] \to \mathbb{R}^{d_1}$ is (pre-trained) and optimal, satisfying $\boldsymbol{v}_\theta((1-t)\boldsymbol{x}_0 + t\boldsymbol{x}_1, t) = \boldsymbol{x}_1 - \boldsymbol{x}_0 \quad \forall \boldsymbol{x}_0 \sim p_{\text{init}}, \boldsymbol{x}_1 \sim p_{\text{prior}}, t \in [0,1]$.*

To further interpret the method, we consider the Assumption 1 that $\boldsymbol{v}_\theta$ is optimally trained such that $\boldsymbol{v}_\theta((1-s)\boldsymbol{x}_0 + s\boldsymbol{x}_1, s) = \boldsymbol{x}_1 - \boldsymbol{x}_0$ for $\boldsymbol{x}_1 \sim p_{\text{prior}}$. If $\boldsymbol{z}$ is itself a sample from $p_{\text{prior}}$, then $\mathcal{L}_{\text{align}}(\boldsymbol{z}; \theta)$ would be (close to) zero. However, when optimizing an arbitrary $\boldsymbol{z}$, especially if it is far from $p_{\text{prior}}$, the $\mathcal{L}_{\text{align}}(\boldsymbol{z}; \theta)$ term can be substantial. Its minimization drives $\boldsymbol{z}$ towards regions of higher plausibility under the learned flow.

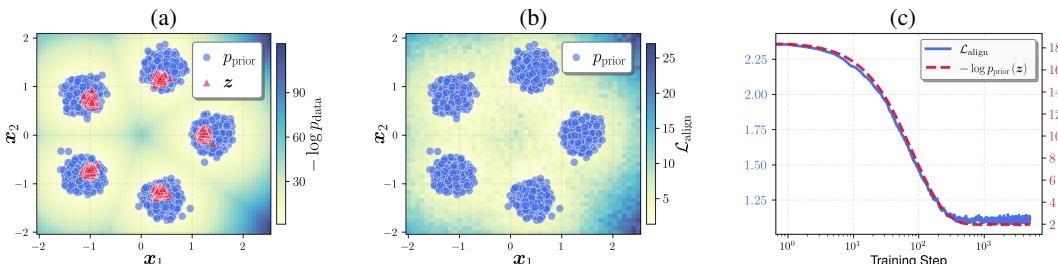

Figure 3: Illustration with a Mixture of Gaussians distribution. (a) Aligned latent variables $\boldsymbol{z}$ (red triangles) concentrate in low negative log-likelihood (NLL) regions of $p_{\text{data}}$ (blue dots; heatmap shows $-\log p_{\text{data}}$). (b) Alignment loss $\mathcal{L}_{\text{align}}$ heatmap mirrors the NLL landscape of $p_{\text{data}}$, with $p_{\text{data}}$ samples in low-$\mathcal{L}_{\text{align}}$ areas. (c) $\mathcal{L}_{\text{align}}$ (blue solid) and $-\log p_{\text{data}}(\boldsymbol{z})$ (red dashed) decline simultaneously in training, showing $\mathcal{L}_{\text{align}}$ serves as a proxy for maximizing the log-likelihood of $\boldsymbol{z}$ under $p_{\text{data}}$.

In practice, directly maximizing $\log p_1^{\boldsymbol{v}_\theta}(\boldsymbol{z})$ via Eq. (4) is computationally demanding, requiring ODE solves and computation of Jacobian traces along these true ODE paths. Maximizing the full ELBO (Eq. (6)) would still require computing $C(\boldsymbol{z})$, which involves trace computations. By focusing on minimizing only $\mathcal{L}_{\text{align}}(\boldsymbol{z}; \theta)$, we adopt a computationally tractable proxy. This objective encourages $\boldsymbol{z}$ to have a high likelihood under $p_1^{\boldsymbol{v}_\theta}(\boldsymbol{z})$ by ensuring consistency with the learned flow dynamics, thereby aligning the distribution of $\boldsymbol{z}$ with the prior distribution $p_{\text{prior}}$ implicitly modeled by $\boldsymbol{v}_\theta$. A more complete proof can be found in Appendix A.

## 5 EXPERIMENTS

This section presents an empirical validation of the proposed alignment method with flow priors. The investigation starts with an illustrative experiment in Sec. 5.1. Subsequently, large-scale experiments are conducted on image generation tasks using the ImageNet dataset, as detailed in Sec. 5.2. In Sec. 5.3, we conduct ablation studies of the proposed method.

### 5.1 TOY EXAMPLES

We present a toy example as an illustrative experiment in a 2D setting. The prior distribution, denoted $p_{\text{prior}}$, is configured as a mixture of five isotropic Gaussians. Following the method outlined in Sec. 4.1, we first train an FM model $\boldsymbol{v}_\theta$ to map a standard Normal $\mathcal{N}(\boldsymbol{0}, \boldsymbol{I})$ to $p_{\text{prior}}$. This FM model $\boldsymbol{v}_\theta$ is implemented by a multi-layer perceptron (MLP) incorporating adaptive layer normalization for time modulation (Peebles & Xie, 2023). Upon completion of training, the parameters $\theta$ of this FM model are frozen. Subsequently, instead of a parameterized model $E_\phi$, we directly initialize a set of learnable 2D variables as $\boldsymbol{z}$ and optimize them by minimizing the alignment loss $\mathcal{L}_{\text{align}}(\boldsymbol{z}; \theta)$.

The results are presented in Fig. 3. Fig. 3 (a) compares the prior distribution $p_{\text{prior}}$ (blue point samples) with the optimized variables $\boldsymbol{z}$ (red triangles). The background visualizes the negative log-likelihood (NLL) of $p_{\text{prior}}$, which is computed analytically. It is evident that $\boldsymbol{z}$ successfully converges to the high log-likelihood regions of $p_{\text{prior}}$. Fig. 3 (b) displays the landscape of the alignment loss $\mathcal{L}_{\text{align}}$, which is estimated numerically with $\boldsymbol{v}_\theta$. The landscape mirrors the NLL surface of $p_{\text{prior}}$ depicted in (a). Samples drawn from $p_{\text{prior}}$ (blue dots) are concentrated in regions where $\mathcal{L}_{\text{align}}$ is low, suggesting that $\mathcal{L}_{\text{align}}$ effectively captures the underlying structure of the prior distribution. Fig. 3 (c) illustrates $\mathcal{L}_{\text{align}}$ (blue solid line) and the true NLL $-\log p_{\text{prior}}(\boldsymbol{z})$ (red dashed line) during the training of $\boldsymbol{z}$. The alignment loss and the NLL exhibit a strong positive correlation, decreasing concomitantly throughout the training process. More detailed toy examples can be found in Appendix B.

### 5.2 IMAGE GENERATION

Prior work has demonstrated that aligning the latent space of AEs with semantic encoders can enhance generative model performance (Chen et al., 2025a; Yao et al., 2025; Chen et al., 2024; Qiu et al., 2025). To validate this observation and further showcase the efficacy of our proposed method, we conduct large-scale image generation experiments on the ImageNet-1K (Deng et al., 2009) dataset at $256 \times 256$ resolution.

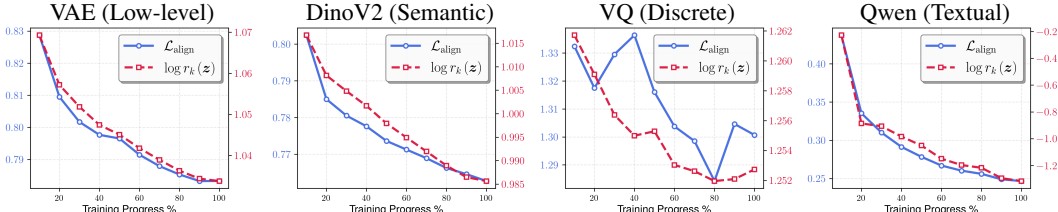

Figure 4: Aligning autoencoders on ImageNet-1K with different prior distributions. The alignment loss $\mathcal{L}_{\text{align}}$ (blue solid) and the $k$-NN distance $\log r_k(\boldsymbol{z})$ (red dashed) are proportional throughout the training. Confirming that $\mathcal{L}_{\text{align}}$ serves as a good proxy for the NLL of the latents under $p_{\text{prior}}$.

**Implementation Details**  Our AE architecture employs two Vision Transformer (ViT)-Large (Dosovitskiy et al., 2021) models, each with 391M parameters, serving as the latent encoder and decoder, respectively. The encoder maps input images to a latent space of 64 tokens, each with dimension 32, striking a balance between reconstruction quality and computational efficiency. We impose *token-level* alignment on the latents. The alignment loss on the latents is set to $\lambda = 0.01$ by default. We also incorporate conventional reconstruction loss, perceptual loss, and adversarial loss on the pixel outputs (Esser et al., 2021; Rombach et al., 2022).

For the prior distribution $p_{\text{prior}}$, we investigate four distinct variants: *low-level* visual features from a VAE (Kingma & Welling, 2014; Esser et al., 2021), continuous *semantic* visual features from DinoV2 (Oquab et al., 2024), *discrete* visual codebook embeddings from LlamaGen VQ (Sun et al., 2024; Van Den Oord et al., 2017), and *textual* embeddings from Qwen (Bai et al., 2023). Their feature dimensions are 32, 768, 8, 896, respectively. The flow-based prior is modeled by a 6-layer MLP with 1024 hidden units, trained for 1 million steps using the AdamW (Loshchilov & Hutter, 2019) optimizer to match Assumption 1. More details can be found in Appendix C.

**Alignment Results**  Analogous to the toy example, we aim to correlate the alignment loss $\mathcal{L}_{\text{align}}$ with the NLL of latents under the prior distribution $p_{\text{prior}}$. Since the NLL is intractable for implicitly defined distributions, we estimate the density using $k$-nearest neighbors ($k = 5$). The probability density $p(\boldsymbol{z})$ at a point $\boldsymbol{z}$ is inversely proportional to the volume of the hypersphere enclosing its $k^{\text{th}}$ nearest neighbor among the prior samples. Consequently, the NLL can be estimated as $-\log p_{\text{prior}}(\boldsymbol{z}) \propto \log r_k(\boldsymbol{z})$ where $r_k(\boldsymbol{z})$ is the Euclidean distance to the $k^{\text{th}}$ neighbor. We use $\log r_k(\boldsymbol{z})$ as our proxy measure for the NLL. We first index the set of prior distribution samples using Faiss (Douze et al., 2024). During the training, we periodically sample 10k points from the latent space and measure the alignment quality by averaging the $\log r_k(\boldsymbol{z})$.

The results are presented in Fig. 4. A strong correlation is observed between the alignment loss $\mathcal{L}_{\text{align}}$ and the $k$-NN distance proxy $\log r_k(\boldsymbol{z})$. The only unstable case is the VQ variant, for which the GAN loss collapses during training due to its low dimension (8-dim), yet the general trend is still consistent. This finding corroborates our conclusion that $\mathcal{L}_{\text{align}}$ serves as an effective proxy for the NLL of the latents under $p_{\text{prior}}$. Crucially, our method captures the underlying structure across diverse prior distributions, spanning different forms (continuous, discrete) and modalities (visual, textual), even when applied to a large-scale dataset like ImageNet and a high-capacity model such as ViT-Large.

**Generation Results**  After demonstrating effective latent space alignment, we investigated its impact on generative model performance. We evaluated both reconstruction and generation capabilities on ImageNet using the MAR-B (Li et al., 2024a) architecture. For MAR-B, we incorporated qk-norm (Dehghani et al., 2023) and replaced the diffusion head with a flow head to ensure stable training. We choose flow-based MAR-B as it does not favor continuous Gaussian-like latent structure like Diffusion models (Song et al., 2021; Dhariwal & Nichol, 2021; Karras et al., 2022; Nichol & Dhariwal, 2021; Rombach et al., 2022) do. To ensure an "apples-to-apples" comparison, configurations and hardware remained identical across all experiments, with the only difference being the specific AE used for each alignment variant.

The results are presented in Tab. 1. Reconstruction performance was measured by rFID (Heusel et al., 2017) and PSNR on the ImageNet validation set. Generation performance was assessed using FID, IS (Salimans et al., 2016), Precision, and Recall on 50k generated samples and the validation set,

Table 1: ImageNet $256 \times 256$ conditional generation using MAR-B. All models are trained and evaluated using identical settings. The CFG scale is tuned for KL and kept the same for others.

| Autoencoder | rFID↓ | PSNR↑ | w/o CFG | | | | w/ CFG | | | |
|---|---|---|---|---|---|---|---|---|---|---|
| | | | FID↓ | IS↑ | Pre.↑ | Rec.↑ | FID↓ | IS↑ | Pre.↑ | Rec.↑ |
| AE | 1.13 | 20.20 | 15.08 | 86.37 | **0.60** | **0.59** | 5.26 | 237.60 | 0.56 | 0.65 |
| KL | 1.65 | 22.59 | 12.94 | 91.86 | 0.60 | 0.58 | 5.29 | 200.85 | 0.57 | 0.65 |
| SoftVQ | **0.61** | 23.00 | 13.30 | 93.40 | 0.60 | 0.57 | 6.09 | 198.53 | **0.58** | 0.61 |
| Low-level (VAE) | 1.22 | 22.31 | 12.04 | 98.66 | 0.56 | 0.57 | 5.02 | 240.03 | 0.56 | 0.62 |
| Semantic (Dino) | 1.26 | 23.07 | **11.47** | 101.74 | 0.59 | 0.59 | **4.87** | **250.38** | 0.54 | 0.67 |
| Discrete (VQ) | 2.99 | 22.32 | 24.63 | 48.17 | 0.55 | 0.53 | 10.04 | 119.64 | 0.47 | 0.65 |
| Textual (Qwen) | 0.85 | **23.12** | 11.89 | **102.23** | 0.55 | 0.57 | 6.56 | 262.89 | 0.49 | **0.69** |

Table 2: Ablation studies on ImageNet $256 \times 256$ for different configurations using autoencoders regularized by textual features (Qwen). We use a shorter training schedule when ablating weight.

(a) Downsampling Methods

| Method | rFID↓ | PSNR↑ | FID↓ | IS↑ |
|---|---|---|---|---|
| Random Proj. | **0.85** | 23.12 | **11.89** | **102.23** |
| Avg. Pooling | 0.94 | 22.98 | 16.06 | 60.37 |
| PCA | 0.87 | **23.14** | 14.95 | 83.59 |

(b) Alignment Loss Weight

| Weight $\lambda$ | rFID↓ | PSNR↑ | FID↓ | IS↑ |
|---|---|---|---|---|
| 0.001 | **0.89** | 22.78 | 17.57 | 75.20 |
| 0.005 | 1.02 | 22.98 | 16.93 | 78.01 |
| 0.01 | 1.31 | **23.12** | 13.67 | 82.13 |
| 0.05 | 1.81 | 21.82 | **12.30** | **92.48** |

both with and without classifier-free guidance (CFG) (Ho & Salimans, 2022). AE does not impose any prior, KL uses a standard Gaussian prior, and SoftVQ performs per-sample alignment with Dino features; all other priors match those used in the alignment experiments. Our key findings are:

*1) Alignment vs. Reconstruction Trade-off:* Latent space alignment typically degrades reconstruction quality (rFID, PSNR) compared to vanilla AEs, as constraints reduce capacity. SoftVQ excels among aligned methods due to its sample-level alignment. *2) Alignment Enhances Generation:* Structured latent spaces improve generative metrics (FID, IS), but complexity is not decisive. Simple features (text embeddings like Qwen) may match the performance of richer visual features (DinoV2). *3) Optimal prior selection is open:* No consensus exists on optimal priors. Low-dimensional discrete features (LlamaGen VQ) underperform, while cross-modal alignment (Qwen text embeddings) demonstrates transferable structural benefits. More discussions can be found in Appendix D.

## 5.3 ABLATION STUDY

**Downsampling Operators** We ablate the downsampling operators in Tab. 2 (a). We adopt the same settings as in Tab. 1 using the model with the textual embeddings (Qwen) as the prior distribution. Despite all being linear downsampling operators, PCA and Avg. Pooling perform worse than Random Projection. We hypothesize that this is because unlike Random Projection which preserves the structure of the data, both PCA and Avg. Pooling are likely to destroy the structure. Avg. Pooling performs especially poorly since it merges close features that are independent from the location.

**Alignment Loss Weight** We apply different strengths of regularization by altering the alignment loss weight $\lambda$ in Tab. 2 (b). As expected, larger weight implies heavier regularization, worse reconstruction, and easier generation. However, heavier regularization limits the generation performance and may even cause the GAN loss to collapse. A trade-off exists between reconstruction and generation when the capacity of the model is limited.

## 6 CONCLUSION

This paper introduced a novel method for aligning learnable latent spaces with arbitrary prior distributions by leveraging pre-trained flow-based generative models as expressive priors. Our approach utilizes a computationally tractable alignment loss, adapted from the flow matching objective, to guide latent variables towards the prior distribution. We theoretically established that minimizing

this alignment loss serves as a proxy for maximizing a variational lower bound on the log-likelihood of the latents under the flow-defined prior. The effectiveness of our method is validated through empirical results, including controlled toy settings and large-scale ImageNet experiments. Ultimately, this work provides a flexible and powerful framework for incorporating rich distributional priors, paving the way for more structured and interpretable representation learning. A *limitation*, and also a promising future direction is that the selection of the optimal prior remains a challenge. While semantic priors are effective for image generation, we posit that no single "silver bullet" prior exists for all tasks; rather, the optimal choice is likely task-specific and needs to be further explored.

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

# A COMPLETE PROOF

## A.1 COMPLETE PROOF FOR PROPOSITION 1

We restate Proposition 1 for clarity and self-containedness.

**Proposition 1.** *Let $\boldsymbol{v}_\theta : \mathbb{R}^{d_1} \times [0,1] \to \mathbb{R}^{d_1}$ be a given velocity field, and $p_{\text{init}}$ be a base distribution. For any $\boldsymbol{z} \in \mathbb{R}^{d_1}$, the log-likelihood $\log p_1^{\boldsymbol{v}_\theta}(\boldsymbol{z})$ of $\boldsymbol{z}$ under the distribution induced by flowing $p_{\text{init}}$ with $\boldsymbol{v}_\theta$ from $t = 0$ to $t = 1$, is lower-bounded as:*

$$\log p_1^{\boldsymbol{v}_\theta}(\boldsymbol{z}) \geq C(\boldsymbol{z}) - \lambda \mathcal{L}_{align}(\boldsymbol{z}; \theta), \tag{12}$$

*where $\lambda > 0$ is a constant, $\mathcal{L}_{align}(\boldsymbol{z}; \theta)$ is defined as*

$$\mathcal{L}_{align}(\boldsymbol{z}; \theta) = \mathbb{E}_{s \sim \mathcal{U}[0,1], \boldsymbol{x}_0 \sim p_{\text{init}}(\boldsymbol{x}_0)} \left[ \|(\boldsymbol{z} - \boldsymbol{x}_0) - \boldsymbol{v}_\theta((1-s)\boldsymbol{x}_0 + s\boldsymbol{z}, s)\|^2 \right], \tag{13}$$

*and $C(\boldsymbol{z})$ is a term dependent on $\boldsymbol{z}$ and $\boldsymbol{v}_\theta$, given by*

$$C(\boldsymbol{z}) = -\mathbb{E}_{s \sim \mathcal{U}[0,1], \boldsymbol{x}_0 \sim p_{\text{init}}} \left[ \text{Tr}(\nabla_{\boldsymbol{z}} \boldsymbol{v}_\theta((1-s)\boldsymbol{x}_0 + s\boldsymbol{z}, s)) \right]. \tag{14}$$

To prove this, we make the following assumptions:

**Assumption 2** (Properties of the Velocity Field). *The velocity field $\boldsymbol{v}_\theta : \mathbb{R}^{d_1} \times [0,1] \to \mathbb{R}^{d_1}$ is continuous in $t$ and Lipschitz continuous in its spatial argument with continuously differentiable components, ensuring that $\nabla_{\boldsymbol{z}} \boldsymbol{v}_\theta(\boldsymbol{z}, s)$ exists and is bounded on compact sets.*

**Assumption 3** (Variational Path Choice). *The variational paths used to construct the ELBO are straight lines $z_s(x_0, z) = (1 - s)x_0 + sz$ for $s \in [0, 1]$, originating from $x_0 \sim p_{\text{init}}$ and terminating at $z$. The velocity of such a path is $\dot{z}_s(x_0, z) = z - x_0$.*

**Assumption 4** (Variational Distribution Choice). *The variational distribution over the initial states $x_0$ conditioned on $z$ is chosen as $q(x_0|z) = p_{\text{init}}(x_0)$.*

**Assumption 5** (Weighting Factor in ELBO). *The time-dependent weighting factor $\lambda_s$ in the general ELBO formulation (Eq. 15 below) is chosen as a positive constant $\lambda_s = \lambda > 0$ for all $s \in [0, 1]$.*

**Remark 1** (Optimality of $v_\theta$). *Sec. 4.3 introduces Assumption 1, which states that $v_\theta$ is optimally pre-trained such that $v_\theta((1 - s)x_0 + sx_1, s) = x_1 - x_0$ for $x_1 \sim p_{\text{data}}$. This assumption is not required for the mathematical validity of the ELBO in Proposition 1 itself, which holds for any $v_\theta$ satisfying Assumption 2. However, Assumption 1 is crucial for interpreting why minimizing $\mathcal{L}_{align}(z; \theta)$ drives $z$ towards $p_{\text{prior}}$, as it implies $\mathcal{L}_{align}(z; \theta) \approx 0$ if $z \sim p_{\text{data}}$.*

*Proof.* The proof is based on a variational approach to lower-bound the log-likelihood in continuous-time generative models. This technique has been established in the literature for Neural ODEs and continuous normalizing flows (Chen et al., 2018; Grathwohl et al., 2019; Liu et al., 2023).

For a sample $z$, we consider a family of paths $z_s(x_0, z)$ parameterized by initial states $x_0$ drawn from a proposal distribution $q(x_0|z)$, where each path starts at $x_0$ and ends at $z$ (i.e., $z_0(x_0, z) = x_0$ and $z_1(x_0, z) = z$). The variational lower bound is derived by considering the path integral formulation of the likelihood. For any such family of paths with velocities $\dot{z}_s(x_0, z)$, the bound takes the form:

$$\log p_1^{v_\theta}(z) \geq \mathbb{E}_{x_0 \sim q(\cdot|z)}\left[ \log p_{\text{init}}(x_0) - \log q(x_0|z) \right.$$

$$- \int_0^1 \lambda_s \|\dot{z}_s(x_0, z) - v_\theta(z_s(x_0, z), s)\|^2 \mathrm{d}s$$

$$\left. - \int_0^1 \mathrm{Tr}(\nabla_{z_s} v_\theta(z_s(x_0, z), s)) \mathrm{d}s \right]. \tag{15}$$

We now apply our specific assumptions:

- By Assumption 3, the paths are $z_s(x_0, z) = (1 - s)x_0 + sz$, and their velocities are $\dot{z}_s(x_0, z) = z - x_0$.

- By Assumption 4, $q(x_0|z) = p_{\text{init}}(x_0)$. This causes the term $\log p_{\text{init}}(x_0) - \log q(x_0|z)$ to vanish.

- By Assumption 5, we set $\lambda_s = \lambda$, a positive constant for all $s \in [0, 1]$.

Substituting these into Eq. 15:

$$\log p_1^{v_\theta}(z) \geq \mathbb{E}_{x_0 \sim p_{\text{init}}}\left[ - \int_0^1 \lambda\|(z - x_0) - v_\theta((1 - s)x_0 + sz, s)\|^2 \mathrm{d}s \right.$$

$$\left. - \int_0^1 \mathrm{Tr}(\nabla_z v_\theta((1 - s)x_0 + sz, s)) \mathrm{d}s \right]. \tag{16}$$

Using the equivalence $\int_0^1 f(s)\mathrm{d}s = \mathbb{E}_{s \sim \mathcal{U}[0,1]}[f(s)]$ for integrable functions $f$, we can rewrite each term. The first term becomes:

$$\mathbb{E}_{x_0 \sim p_{\text{init}}}\left[ -\lambda \int_0^1 \|(z - x_0) - v_\theta((1 - s)x_0 + sz, s)\|^2 \mathrm{d}s \right]$$

$$= -\lambda \mathcal{L}_{\text{align}}(z; \theta),$$

using the definition of $\mathcal{L}_{\text{align}}(z; \theta)$ from Eq. 13. The second term becomes:

$$\mathbb{E}_{x_0 \sim p_{\text{init}}}\left[ - \int_0^1 \mathrm{Tr}(\nabla_z v_\theta((1 - s)x_0 + sz, s)) \mathrm{d}s \right] = C(z),$$

using the definition of $C(\boldsymbol{z})$ from Eq. 14. Combining these, the ELBO becomes:

$$\log p_1^{\boldsymbol{v}_\theta}(\boldsymbol{z}) \geq C(\boldsymbol{z}) - \lambda \mathcal{L}_{\text{align}}(\boldsymbol{z}; \theta). \tag{17}$$

This completes the proof of Proposition 1. Our paper uses $\lambda = 1$ for simplicity, yielding the bound $C(\boldsymbol{z}) - \mathcal{L}_{\text{align}}(\boldsymbol{z}; \theta)$. $\qquad\square$

## A.2 Rigorous Analysis of $C(\boldsymbol{z})$

We now provide a rigorous analysis of the term $C(\boldsymbol{z})$ in the ELBO and establish conditions under which minimizing $\mathcal{L}_{\text{align}}(\boldsymbol{z}; \theta)$ leads to favorable behavior of $C(\boldsymbol{z})$.

### A.2.1 Geometric Interpretation of $C(\boldsymbol{z})$

The term $C(\boldsymbol{z})$ represents the negative expected divergence of the velocity field $\boldsymbol{v}_\theta$ along straight-line variational paths from $\boldsymbol{x}_0 \sim p_{\text{init}}$ to $\boldsymbol{z}$:

$$C(\boldsymbol{z}) = -\mathbb{E}_{s \sim \mathcal{U}[0,1], \boldsymbol{x}_0 \sim p_{\text{init}}} \left[ \text{Tr}(\nabla_{\boldsymbol{z}} \boldsymbol{v}_\theta((1-s)\boldsymbol{x}_0 + s\boldsymbol{z}, s)) \right]. \tag{18}$$

To understand its role, recall that in the exact likelihood computation for a flow model, we have:

$$\log p_1^{\boldsymbol{v}_\theta}(\boldsymbol{z}) = \log p_{\text{init}}(\boldsymbol{x}_0^*(\boldsymbol{z})) - \int_0^1 \text{Tr}(\nabla_{\boldsymbol{x}} \boldsymbol{v}_\theta(\boldsymbol{x}_s^*(\boldsymbol{z}), s)) \mathrm{d}s, \tag{19}$$

where $\boldsymbol{x}_s^*(\boldsymbol{z})$ is the unique ODE trajectory satisfying $\dot{\boldsymbol{x}}_s^* = \boldsymbol{v}_\theta(\boldsymbol{x}_s^*, s)$ with $\boldsymbol{x}_1^*(\boldsymbol{z}) = \boldsymbol{z}$. The divergence integral measures the logarithmic volume change induced by the flow.

Our variational bound approximates this exact computation by averaging over straight-line paths rather than the true ODE trajectory. The quality of this approximation depends on how well the straight paths approximate the true flow geometry.

### A.2.2 Relationship Between $C(\boldsymbol{z})$ and Distributional Alignment

We establish the key relationship between $C(\boldsymbol{z})$ and the alignment quality measured by $\mathcal{L}_{\text{align}}(\boldsymbol{z}; \theta)$.

**Lemma 1** (Consistency of Variational Paths)**.** *Under Assumption 1 (optimal $\boldsymbol{v}_\theta$), for $\boldsymbol{z} \sim p_{\text{data}}$, the straight-line variational paths $\boldsymbol{z}_s = (1-s)\boldsymbol{x}_0 + s\boldsymbol{z}$ satisfy:*

$$\mathbb{E}_{\boldsymbol{x}_0 \sim p_{\text{init}}} \left[ \|(\boldsymbol{z} - \boldsymbol{x}_0) - \boldsymbol{v}_\theta(\boldsymbol{z}_s, s)\|^2 \right] = 0 \quad \forall s \in [0, 1]. \tag{20}$$

*Consequently, $\mathcal{L}_{align}(\boldsymbol{z}; \theta) = 0$ when $\boldsymbol{z} \sim p_{\text{data}}$.*

*Proof.* By Assumption 1, for $\boldsymbol{z} \sim p_{\text{data}}$ and $\boldsymbol{x}_0 \sim p_{\text{init}}$, we have:

$$\boldsymbol{v}_\theta((1-s)\boldsymbol{x}_0 + s\boldsymbol{z}, s) = \boldsymbol{z} - \boldsymbol{x}_0.$$

Therefore, $\|(\boldsymbol{z} - \boldsymbol{x}_0) - \boldsymbol{v}_\theta((1-s)\boldsymbol{x}_0 + s\boldsymbol{z}, s)\|^2 = 0$ for all $\boldsymbol{x}_0$ and $s$, which implies the result. $\qquad\square$

**Theorem 1** (Monotonic Behavior of the ELBO)**.** *Consider two points $\boldsymbol{z}_1, \boldsymbol{z}_2 \in \mathbb{R}^{d_1}$ such that $\mathcal{L}_{align}(\boldsymbol{z}_1; \theta) > \mathcal{L}_{align}(\boldsymbol{z}_2; \theta)$. If the velocity field $\boldsymbol{v}_\theta$ is L-Lipschitz in its spatial argument and satisfies Assumption 1, then:*

$$|C(\boldsymbol{z}_1) - C(\boldsymbol{z}_2)| \leq L \cdot d_1 \cdot \tfrac{1}{2} \|\boldsymbol{z}_1 - \boldsymbol{z}_2\|, \tag{21}$$

*where the factor $\frac{1}{2}$ comes from $\mathbb{E}_{s \sim \mathcal{U}[0,1]}[s] = \frac{1}{2}$. Moreover, if $\mathcal{L}_{align}(\boldsymbol{z}_1; \theta) - \mathcal{L}_{align}(\boldsymbol{z}_2; \theta) > \frac{L \cdot d_1}{2} \cdot \|\boldsymbol{z}_1 - \boldsymbol{z}_2\|$, then:*

$$\log p_1^{\boldsymbol{v}_\theta}(\boldsymbol{z}_2) - \log p_1^{\boldsymbol{v}_\theta}(\boldsymbol{z}_1) > 0. \tag{22}$$

*Proof.* From the definition of $C(\boldsymbol{z})$ in Eq. 14:

$$\begin{aligned} C(\boldsymbol{z}_1) - C(\boldsymbol{z}_2) = &-\mathbb{E}_{s, \boldsymbol{x}_0} \left[ \text{Tr}(\nabla_{\boldsymbol{z}} \boldsymbol{v}_\theta((1-s)\boldsymbol{x}_0 + s\boldsymbol{z}_1, s)) \right] \\ &+ \mathbb{E}_{s, \boldsymbol{x}_0} \left[ \text{Tr}(\nabla_{\boldsymbol{z}} \boldsymbol{v}_\theta((1-s)\boldsymbol{x}_0 + s\boldsymbol{z}_2, s)) \right]. \end{aligned} \tag{23}$$

By the Lipschitz continuity of $\boldsymbol{v}_\theta$, its Jacobian $\nabla_{\boldsymbol{z}}\boldsymbol{v}_\theta(\boldsymbol{z},s)$ has bounded operator norm $\|\nabla_{\boldsymbol{z}}\boldsymbol{v}_\theta(\boldsymbol{z},s)\|_{op} \leq L$. Therefore:

$$\begin{aligned}|\mathrm{Tr}(\nabla_{\boldsymbol{z}}\boldsymbol{v}_\theta(\boldsymbol{u}_1,s)) - \mathrm{Tr}(\nabla_{\boldsymbol{z}}\boldsymbol{v}_\theta(\boldsymbol{u}_2,s))| &\leq d_1 \cdot \|\nabla_{\boldsymbol{z}}\boldsymbol{v}_\theta(\boldsymbol{u}_1,s) - \nabla_{\boldsymbol{z}}\boldsymbol{v}_\theta(\boldsymbol{u}_2,s)\|_{op} \\ &\leq d_1 \cdot L \cdot \|\boldsymbol{u}_1 - \boldsymbol{u}_2\|,\end{aligned} \tag{24}$$

where we set $\boldsymbol{u}_1 = (1-s)\boldsymbol{x}_0 + s\boldsymbol{z}_1$ and $\boldsymbol{u}_2 = (1-s)\boldsymbol{x}_0 + s\boldsymbol{z}_2$, so that

$$\|\boldsymbol{u}_1 - \boldsymbol{u}_2\| = s\|\boldsymbol{z}_1 - \boldsymbol{z}_2\|.$$

Taking expectations over $s \sim \mathcal{U}[0,1]$ yields Eq. 21.

For the second part, using the ELBO bound from Proposition 1:

$$\begin{aligned}\log p_1^{\boldsymbol{v}_\theta}(\boldsymbol{z}_2) - \log p_1^{\boldsymbol{v}_\theta}(\boldsymbol{z}_1) &\geq [C(\boldsymbol{z}_2) - \mathcal{L}_{\mathrm{align}}(\boldsymbol{z}_2;\theta)] - [C(\boldsymbol{z}_1) - \mathcal{L}_{\mathrm{align}}(\boldsymbol{z}_1;\theta)] \\ &= [C(\boldsymbol{z}_2) - C(\boldsymbol{z}_1)] + [\mathcal{L}_{\mathrm{align}}(\boldsymbol{z}_1;\theta) - \mathcal{L}_{\mathrm{align}}(\boldsymbol{z}_2;\theta)].\end{aligned} \tag{25}$$

Using the bound on $|C(\boldsymbol{z}_1) - C(\boldsymbol{z}_2)|$ and the condition on $\mathcal{L}_{\mathrm{align}}(\boldsymbol{z}_1;\theta) - \mathcal{L}_{\mathrm{align}}(\boldsymbol{z}_2;\theta)$, the result follows. $\square$

### A.2.3 ANALYSIS OF THE IDEALIZED CASE

We address the mathematical singularity that arises in the idealized rectified flow case where $\boldsymbol{v}_\theta$ has the exact form $\boldsymbol{v}_\theta(\boldsymbol{z},s) = \boldsymbol{z}/s$ for $s > 0$.

**Proposition 3** (Regularization by Neural Network Parameterization). *Let $\boldsymbol{v}_\theta$ be parameterized by a neural network with bounded weights. Then there exists a constant $M > 0$ such that:*

$$|\mathrm{Tr}(\nabla_{\boldsymbol{z}}\boldsymbol{v}_\theta(\boldsymbol{z},s))| \leq M \quad \forall \boldsymbol{z} \in \text{compact sets}, \, s \in [\epsilon, 1] \tag{26}$$

*for any $\epsilon > 0$. Consequently, $C(\boldsymbol{z})$ is well-defined and finite.*

*Proof.* Neural networks with bounded parameters have Lipschitz continuous components. The Jacobian $\nabla_{\boldsymbol{z}}\boldsymbol{v}_\theta(\boldsymbol{z},s)$ inherits this boundedness on compact sets, preventing the $1/s$ singularity from occurring exactly. The trace is therefore bounded, ensuring $C(\boldsymbol{z})$ remains finite. $\square$

### A.2.4 PRACTICAL IMPLICATIONS AND OPTIMIZATION STRATEGY

Our analysis establishes that:

**1. Consistency Principle:** When $\boldsymbol{z} \sim p_{\mathrm{data}}$, both $\mathcal{L}_{\mathrm{align}}(\boldsymbol{z};\theta) = 0$ and $C(\boldsymbol{z})$ takes on the value appropriate for samples from the prior distribution.

**2. Monotonicity Property:** Theorem 1 shows that sufficiently large reductions in $\mathcal{L}_{\mathrm{align}}(\boldsymbol{z};\theta)$ guarantee improvements in the ELBO lower bound, even accounting for changes in $C(\boldsymbol{z})$.

**3. Computational Tractability:** While computing $C(\boldsymbol{z})$ requires evaluating Jacobian traces, minimizing only $\mathcal{L}_{\mathrm{align}}(\boldsymbol{z};\theta)$ provides a computationally efficient proxy that, by Theorem 1, leads to ELBO improvements under reasonable conditions.

**4. Robustness:** Proposition 3 ensures that practical neural network implementations avoid the theoretical singularities, making the method stable in practice.

This analysis demonstrates that minimizing $\mathcal{L}_{\mathrm{align}}(\boldsymbol{z};\theta)$ is not merely heuristic but has solid theoretical foundation as a strategy for maximizing the variational lower bound on $\log p_1^{\boldsymbol{v}_\theta}(\boldsymbol{z})$.

## A.3 THE SIGNIFICANCE OF ASSUMPTIONS

The derivation of Proposition 1 and its interpretation rely on several assumptions, as listed in Sec. A.1. In this section, we discuss the significance of each assumption.

**Assumption 2 (Properties of the Velocity Field):** Lipschitz continuity of $\boldsymbol{v}_\theta$ in its spatial argument ensures that the ODE $\dot{\boldsymbol{z}}_t = \boldsymbol{v}_\theta(\boldsymbol{z}_t,t)$ has unique solutions, fundamental for defining $p_1^{\boldsymbol{v}_\theta}(\boldsymbol{z})$. Differentiability is required for the Jacobian $\nabla_{\boldsymbol{z}}\boldsymbol{v}_\theta$ to exist, and thus for the divergence term $\mathrm{Tr}(\nabla_{\boldsymbol{z}}\boldsymbol{v}_\theta)$ in the ELBO to be well-defined. These are standard regularity conditions for flow-based models. Without them, the ELBO formulation would be ill-defined.

**Assumption 3 (Variational Path Choice):** The choice of straight-line paths $\boldsymbol{z}_s(\boldsymbol{x}_0, \boldsymbol{z}) = (1-s)\boldsymbol{x}_0 + s\boldsymbol{z}$ is a specific variational decision. This leads to the path velocity $\dot{\boldsymbol{z}}_s = \boldsymbol{z} - \boldsymbol{x}_0$, which is key to the definition of $\mathcal{L}_{\text{align}}(\boldsymbol{z}; \theta)$. This assumption is thus crucial for the specific form of $\mathcal{L}_{\text{align}}$ used.

**Assumption 4 (Variational Distribution Choice):** Setting $q(\boldsymbol{x}_0|\boldsymbol{z}) = p_{\text{init}}(\boldsymbol{x}_0)$ greatly simplifies the ELBO by causing the term $\log p_{\text{init}}(\boldsymbol{x}_0) - \log q(\boldsymbol{x}_0|\boldsymbol{z})$ to vanish. This common choice implies the ELBO considers paths originating from the prior, without inferring a specific $\boldsymbol{x}_0$ for each $\boldsymbol{z}$. While simplifying, this choice affects the ELBO's tightness.

**Assumption 5 (Weighting Factor in ELBO):** Choosing $\lambda_s = \lambda$ makes the loss term in the ELBO directly correspond to $\mathcal{L}_{\text{align}}$. A time-dependent $\lambda_s > 0$ is also valid and could yield a tighter bound or differentially weight errors across time $s$. The constant $\lambda$ ensures a direct link to the standard L2 norm in $\mathcal{L}_{\text{align}}$. This choice affects the ELBO's value but not its validity as a lower bound.

**Assumption 1 (Optimality of $\boldsymbol{v}_\theta$):** As detailed in Remark 1 of Sec. A.1, this assumption is not necessary for the mathematical derivation of Proposition 1 itself; the ELBO inequality holds for any $\boldsymbol{v}_\theta$ satisfying Assumption 2. However, Assumption 1 is paramount for the *interpretation* and *effectiveness* of minimizing $\mathcal{L}_{\text{align}}(\boldsymbol{z}; \theta)$ as a strategy to align $\boldsymbol{z}$ with $p_{\text{prior}}$. If $\boldsymbol{v}_\theta$ is optimal as defined, then $\mathcal{L}_{\text{align}}(\boldsymbol{z}; \theta)$ will be minimized ideally to zero when $\boldsymbol{z}$ is drawn from $p_{\text{prior}}$. Consequently, minimizing this loss for $\boldsymbol{z}$ encourages $\boldsymbol{z}$ to conform to $p_{\text{prior}}$.

In essence, Assumptions 2 through 5 are primarily structural, defining the specific ELBO being analyzed. They ensure the bound is well-defined and takes the presented form. Assumption 1 concerning the optimality of $\boldsymbol{v}_\theta$ is interpretative, providing the rationale for why minimizing a component of this ELBO ($\mathcal{L}_{\text{align}}$) is a meaningful objective for achieving distributional alignment. The overall conclusion that minimizing $\mathcal{L}_{\text{align}}$ serves as a proxy for maximizing a log-likelihood lower bound relies on these assumptions.

# B  ADDITIONAL TOY EXAMPLES

To further demonstrate the effectiveness of our proposed method, we present additional toy examples with diverse prior distributions $p_{\text{prior}}$: a Grid of Gaussians, Two Moons, Concentric Rings, a Spiral, and a Swiss Roll. For each of these distributions, following the visualization style of Fig. 3, we illustrate: (a) The optimized variables $\boldsymbol{z}$ (red triangles) and samples from $p_{\text{prior}}$ (blue dots), overlaid on the negative log-likelihood (NLL) landscape of $p_{\text{prior}}$ (background heatmap showing $-\log p_{\text{prior}}(\cdot)$). (b) The landscape of the alignment loss $\mathcal{L}_{\text{align}}$ (background heatmap), with samples from $p_{\text{prior}}$ (blue dots). (c) The evolution of $\mathcal{L}_{\text{align}}(\boldsymbol{z}; \theta)$ (blue solid line) and the true NLL $-\log p_{\text{prior}}(\boldsymbol{z})$ (red dashed line) during the optimization of $\boldsymbol{z}$.

For the Grid of Gaussians, which is also a mixture of Gaussians, the NLL $-\log p_{\text{prior}}(\boldsymbol{z})$ is computed analytically. For the other distributions (Two Moons, Concentric Rings, Spiral, and Swiss Roll), where an analytical form for $p_{\text{prior}}$ is not readily available, we estimate the NLL using Kernel Density Estimation (KDE). This estimation is based on $N = 100,000$ samples drawn from the respective $p_{\text{prior}}$ and employs a Gaussian kernel with a bandwidth of $h = 0.1$. The probability density $\hat{p}_{\text{prior}}(\mathbf{x})$ at a point $\mathbf{x}$ is estimated as:

$$\hat{p}_{\text{prior}}(\mathbf{x}) = \frac{1}{Nh^d} \sum_{i=1}^{N} K\left(\frac{\mathbf{x} - \mathbf{x}_i}{h}\right), \tag{27}$$

where $\mathbf{x}_i$ are the $N$ samples drawn from $p_{\text{prior}}$, $d$ is the dimensionality (here, $d = 2$), and $K(\cdot)$ is the Gaussian kernel function. The NLL for an optimized variable $\boldsymbol{z}$ is then approximated by $-\log(\hat{p}_{\text{prior}}(\boldsymbol{z}))$. This provides an empirical measure of how well $\boldsymbol{z}$ aligns with the prior distribution as estimated by KDE.

The results for these additional toy examples are comprehensively presented in Fig. 5. Each row in this figure corresponds to one of the five prior distributions. The left column (a,d,g,j,m) shows that the optimized variables $\boldsymbol{z}$ (red triangles) successfully converge to the high-density (low-NLL) regions of $p_{\text{prior}}$. The middle column (b,e,h,k,n) demonstrates that the landscape of our alignment loss $\mathcal{L}_{\text{align}}$ closely mirrors the NLL surface of $p_{\text{prior}}$, with true data samples (blue dots) residing in low-loss areas. The right column (c,f,i,l,o) confirms the strong positive correlation between $\mathcal{L}_{\text{align}}$ and the NLL of $\boldsymbol{z}$, as both decrease concomitantly during optimization. Furthermore, Fig. 6 visualizes the

optimization trajectory of $\boldsymbol{z}$ for the initial mixture of Gaussians (from Sec. 5.1) alongside the five additional toy distributions. These sequential snapshots illustrate how minimizing $\mathcal{L}_{\text{align}}$ effectively steers the variables $\boldsymbol{z}$ from their initialization towards the intricate structures of the prior distributions, reinforcing the robustness and efficacy of our alignment loss.

## C  IMPLEMENTATION DETAILS

### C.1  IMPLEMENTATION DETAILS OF THE TOY EXAMPLE

The primary toy example, illustrated in Figure 3, utilizes a 2D Mixture of Gaussians (MoG) as the prior distribution $p_{\text{prior}}(\boldsymbol{x})$. This MoG distribution consists of 5 components, each with an isotropic standard deviation of 0.3. The means of these Gaussian components are distributed evenly on a circle of radius 3.0. Prior to model training, samples drawn from this MoG distribution are normalized by dividing by their standard deviation, which is empirically computed from a large batch of 10 million samples. In addition to the MoG, our toy experiments also encompassed other 2D synthetic distributions, including Spiral, Moons, Concentric Rings, Swiss Roll, and Grid of Gaussians, to demonstrate the versatility of our approach. The general setup for the flow model and learnable latents applies across these various distributions.

The conditional flow model, denoted $v_\phi(\boldsymbol{x}, t)$, is implemented using a MLP with AdaLN. This network has 2 input channels, 2 output channels, a hidden dimensionality of 512, and incorporates 4 residual blocks. The flow model is trained for $1m$ steps using the AdamW optimizer (beta values of (0.9, 0.999) and no weight decay) with a constant learning rate of $1 \times 10^{-4}$, and a batch size of 256.

A set of 1,000 learnable latent variables $\{\boldsymbol{z}_i\}$ is initialized by sampling from a standard normal distribution $\mathcal{N}(\boldsymbol{0}, \boldsymbol{I})$. These latents are then optimized to align with the prior distribution $p_{\text{prior}}$ by minimizing the alignment loss $\mathcal{L}_{\text{align}}$. This alignment training phase also employs the Adam optimizer (betas=(0.9, 0.999), no weight decay), with a learning rate of $1 \times 10^{-2}$, and runs for 5,000 steps.

Table 3: Training Hyperparameters

| Hyperparameter | Flow | Autoencoder | MAR |
|---|---|---|---|
| Global Batch Size | | 256 | |
| Steps | $1m$ | $50k$ | $250k$ |
| Optimizer | | AdamW | |
| Base Learning Rate | | $1.0 \times 10^{-4}$ | |
| LR Scheduler | Cosine | Cosine | Constant |
| Warmup Steps | 2.5k | 2.5k | 62.5k |
| Adam $\beta_1$ | | 0.9 | |
| Adam $\beta_2$ | 0.95 | 0.95 | 0.999 |
| Weight Decay | $1.0 \times 10^{-4}$ | $1.0 \times 10^{-4}$ | 0.02 |
| Max Grad Norm | | 1.0 | |
| Mixed Precision | | BF16 | |
| EMA Rate | | 0.9999 | |

### C.2  IMPLEMENTATION DETAILS OF THE FLOW MODEL

The flow model $\boldsymbol{v}_\theta(\boldsymbol{z}, t) : \mathbb{R}^{d_1} \times [0, 1] \to \mathbb{R}^{d_1}$ is implemented as a multi-layer perceptron (MLP) with 6 layers and 1024 hidden units per layer. The network employs GELU activation functions and incorporates time modulation through adaptive layer normalization (AdaLN) to handle the temporal dimension $t$. When dimension mismatch occurs between the latent space dimension $d_1$ and prior feature space dimension $d_2$, fixed linear projection layers are applied to map prior features to the appropriate dimension. These projection matrices are initialized with Gaussian weights scaled by $1/\sqrt{d_2}$ and remain frozen during training.

The flow model is trained using the flow matching objective on the prior distribution $p_{\text{prior}}$ for 1 million steps. During training, the model learns to predict velocity fields that transport samples from a standard Gaussian base distribution $\mathcal{N}(\boldsymbol{0}, \boldsymbol{I})$ to the prior distribution along straight-line interpolation

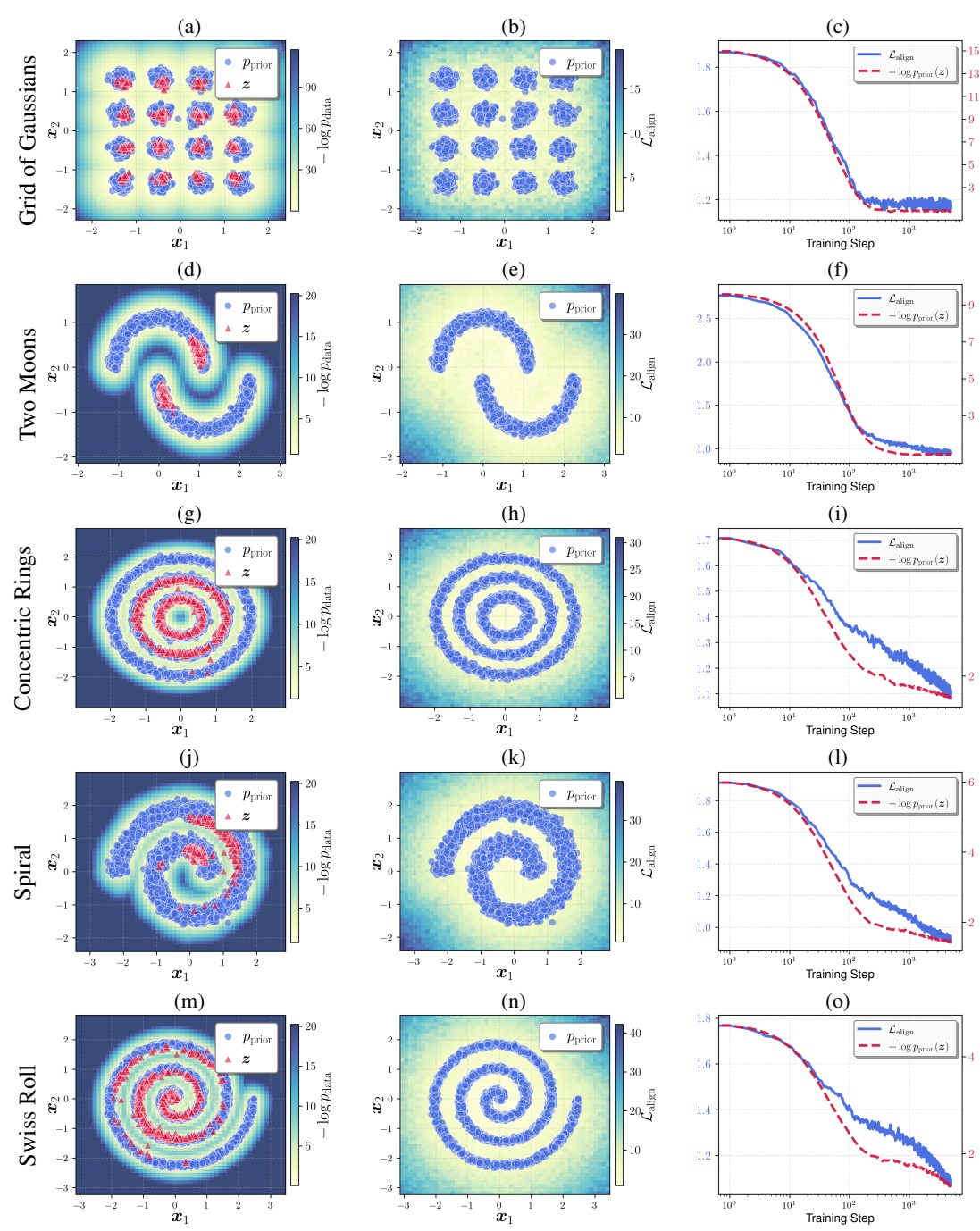

Figure 5: Further illustrations of our method's performance on various 2D toy examples. Each row corresponds to a different prior distribution $p_{\text{prior}}$ (Grid of Gaussians, Two Moons, Concentric Rings, Spiral, and Swiss Roll). **Left column (a,d,g,j,m):** Optimized variables $z$ (red triangles) and samples from $p_{\text{prior}}$ (blue dots). The background heatmap visualizes the negative log-likelihood (NLL) $-\log p_{\text{prior}}(\cdot)$, with $z$ converging to low-NLL (high-density) regions. **Middle column (b,e,h,k,n):** The landscape of the alignment loss $\mathcal{L}_{\text{align}}$ (heatmap) with $p_{\text{prior}}$ samples (blue dots). This landscape mirrors the NLL surface, and $p_{\text{prior}}$ samples are concentrated in areas of low $\mathcal{L}_{\text{align}}$. **Right column (c,f,i,l,o):** Training curves for $\mathcal{L}_{\text{align}}(z;\theta)$ (blue solid line) and NLL $-\log p_{\text{prior}}(z)$ (red dashed line). Their strong positive correlation and concurrent decrease during optimization demonstrate that $\mathcal{L}_{\text{align}}$ effectively serves as a proxy for maximizing the log-likelihood of $z$ under $p_{\text{prior}}$.

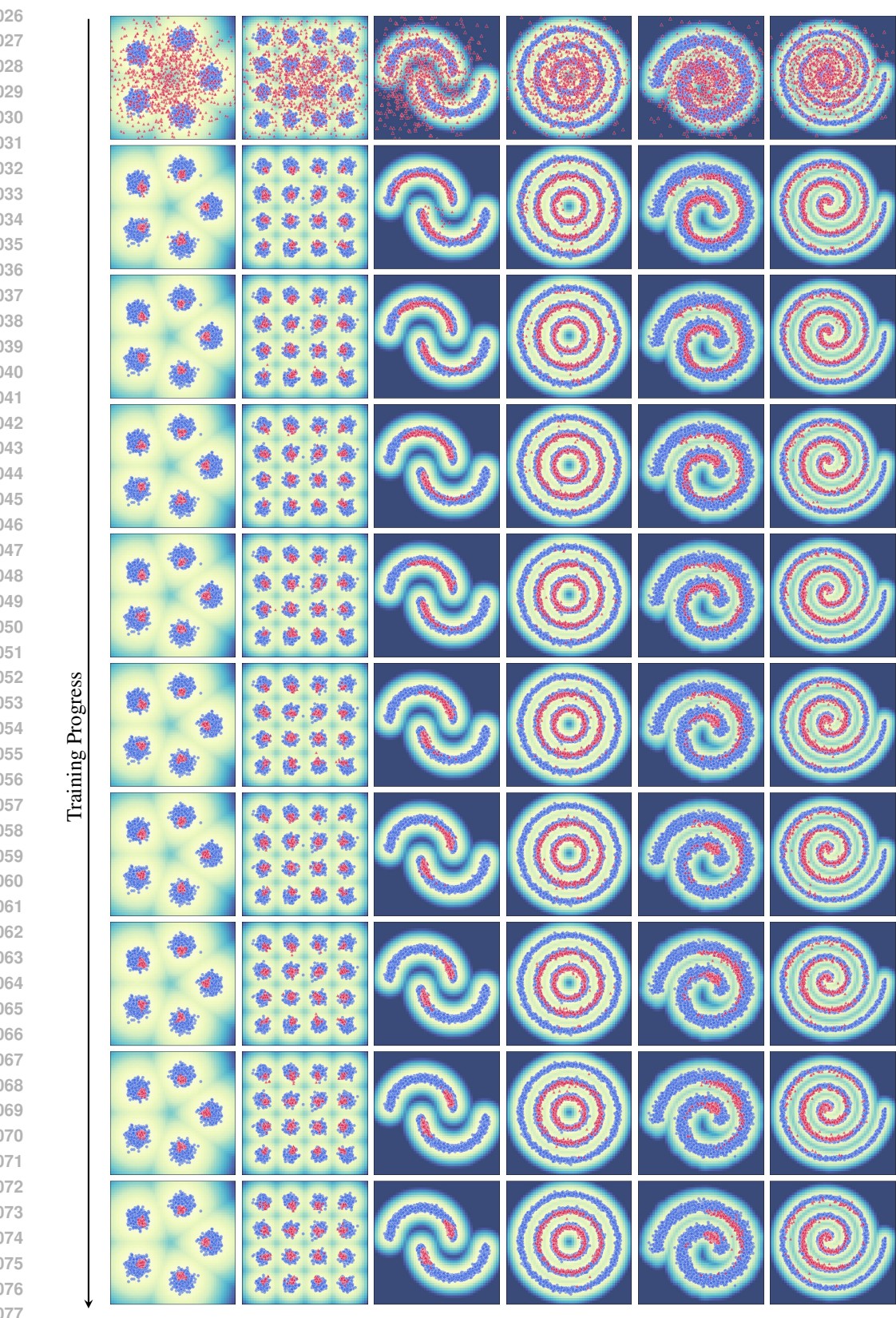

Figure 6: Evolution of the optimized variables $z$ (red triangles) during training across various toy examples. Each column represents a prior distribution $p_{\text{prior}}$. The training progress demonstrates how minimizing $\mathcal{L}_{\text{align}}$ guides $z$ to converge towards low-NLL (high-density) regions of $p_{\text{prior}}$.

paths. The training employs mixed precision (BF16) with gradient clipping and exponential moving averages (EMA). Upon completion of training, the flow model parameters $\theta$ are frozen and used for subsequent latent space alignment. Detailed hyperparameters are provided in Table 3.

## C.3 Implementation Details of Autoencoders

Our autoencoder architecture follows the SoftVQ design, which employs Vision Transformer (ViT) based encoder and decoder networks. The encoder utilizes a ViT-Large model with patch size 14 from DINOv2 (Oquab et al., 2024), initialized with pre-trained weights and fine-tuned with full parameter updates during training. The decoder employs the same ViT-Large architecture but is initialized randomly without pre-trained weights.

The training process utilizes adversarial loss with a DINOv2-based discriminator, incorporating patch-based adversarial training with hinge loss formulation. Perceptual loss is applied using VGG features with a warmup period of $10k$ steps. The model is trained for $50k$ steps with cosine learning rate scheduling and exponential moving averages for stable training dynamics. Unlike SoftVQ, we do not employ the sample-level alignment loss (i.e., REPA loss), making our method more general and efficient. Detailed hyperparameters are provided in Table 3.

We followed the SoftVQ implementation as closely as possible. While we can reproduce almost identical reconstruction results, our tokenizer doesn't quite match the generation performance of the released pre-trained model, even after significant effort to optimize it. We believe this gap comes from differences in the cleaned-up code and the specific hardware we used for training. To keep things fair and validate the effectiveness of our method, we conduct all experiments on *the same hardware with identical training settings*.

All experiments are run on a single node with 8 GPUs. Training the MLP flow prior on features for 1M steps takes only a few hours of wall-clock time. In comparison, training the autoencoder for $50k$ steps with reconstruction, adversarial, perceptual, and alignment losses requires 50 hours, and training the MAR-B conditional generator requires 70 hours. In total, the autoencoder and generative model together take about 5 days to train, whereas the one-time cost of the flow prior is negligible at this scale. During autoencoder training, the alignment term requires only a single forward pass through the frozen MLP per iteration and did not noticeably change the overall training throughput relative to a KL-regularized baseline.

## C.4 Implementation Details of MAR

We follow the original MAR-B implementation with several key modifications. We incorporate qk-norm in the attention mechanism and replace the diffusion head with a flow-based head trained using per-token flow matching loss. The original SD-KL-16 autoencoder is replaced with our trained autoencoders, applying input normalization with scaling factor 1.7052 estimated from sample batches.

Our model uses MAR-B architecture with $256 \times 256$ input images. The flow-based MLP head features adaptive layer normalization with 6 layers and 1024 hidden units per layer, identical to the original diffusion implementation. The model processes sequences of length 64 corresponding to our 64-token latent representation. More training details are provided in Table 3.

For inference, we employ an Euler sampler with 100 steps for the flow-based generation. The autoregressive sampling is limited to 64 steps. Generation uses batch size 256 and produces 50,000 images for evaluation. All evaluations use the standard toolkit from guided diffusion with FID and IS metrics computed at regular intervals during training.

## D Additional Discussions for the Experiments

Here we provide additional discussions and analysis for the experiments presented in Section 5.2.

**Does Johnson-Lindenstrauss Lemma Really Hold?** While the Johnson-Lindenstrauss (JL) lemma theoretically guarantees that random projections preserve distances with high probability, our experimental setup violates its conditions due to the large sample size relative to the prior dimension. However, our results demonstrate that random projections can still preserve distributional structure

to a sufficient extent for effective alignment. In our ablation study with Tab. 2a, random projection achieves the best performance with FID of 11.89 and IS of 102.23, significantly outperforming PCA (FID: 14.95, IS: 83.59) and average pooling (FID: 16.06, IS: 60.37). This suggests that the structure-preserving properties of random projections, even when the JL lemma doesn't strictly hold, are more beneficial than the variance-maximizing properties of PCA or the spatial averaging of pooling operations.

**Continuous or Discrete?**   Our method demonstrates robustness across both continuous and discrete prior distributions. Continuous semantic features from DinoV2 achieve the best generation performance among all variants in Tab. 1, and the discrete textual features from Qwen also achieve effective performance. In contrast, discrete VQ features perform poorly, likely due to structural limitations imposed by low dimensionality (8-dim). The collapse observed in discrete VQ experiments during training can be attributed to the insufficient capacity of the low-dimensional latent space to capture the complexity of ImageNet data while simultaneously satisfying the alignment constraint.

**Why Textual Features Work?**   The surprising effectiveness of textual embeddings (Qwen) for visual generation warrants deeper analysis. Despite being trained on text data, Qwen embeddings achieve competitive generation performance (FID: 11.89 without CFG) and the best PSNR (23.12) among aligned methods. This suggests that high-quality textual representations capture abstract semantic structures that are transferable across modalities. The 896-dimensional Qwen embeddings provide a rich semantic space that can effectively constrain the visual latent space without being overly restrictive. This cross-modal transferability indicates that the structural benefits of alignment are not limited to within-modality features.

**Is Generation Loss a Good Indicator?**   The training loss in generation of our aligned autoencoders is significantly lower than other models. However, we observe that lower training losses do not necessarily translate to better generation results, even for flow-based models where loss is proven to be a direct indicator for generation performance. This paradox can be attributed to the simplification of the latent space under strong alignment constraints. While simplified latent spaces are easier for generative models to sample from (hence lower training losses), they may sacrifice the diversity and fine-grained details necessary for high-quality generation. This suggests that generation quality depends not only on the ease of modeling the latent distribution but also on the expressiveness and diversity preserved in the aligned space.

**How to Select the Prior?**   The optimal choice of prior distribution remains an open research question. Our experiments suggest several guidelines: (1) Higher dimensionality generally enables better performance, as evidenced by the poor performance of 8-dimensional VQ features compared to higher-dimensional alternatives. (2) Semantic richness matters, but not necessarily complexity—simple textual features can match sophisticated visual features. (3) The structural properties of the prior distribution (e.g., smoothness, cluster separation) may be more important than its semantic content for generation quality.

**Does the model align even without alignment loss?**   To empirically verify that, without our alignment loss, the model has no signal from the prior to guide it, we plot the k-NN distance metric $\log r_k(z)$ for an autoencoder with a Dino prior target but without the alignment objective. As expected, the latent space does not align with the prior and, in fact, the alignment metric degrades over the course of training, in stark contrast to our full method.

# E   LLM USAGE DECLARATION

This paper makes use of large language models (LLMs) exclusively for non-substantive assistance, including grammar correction, text formatting, LaTeX syntax support, and proofreading. All research ideas, mathematical formulations, experimental design, analysis, and conclusions are the original work of the authors.

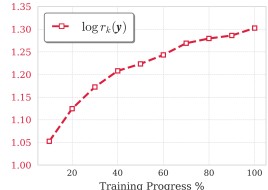

Figure 7: Alignment behavior of an autoencoder with a Dino prior target but without the proposed alignment loss. The k-NN distance metric $\log r_k(z)$ fails to improve and instead drifts away from the prior, indicating that the latent space does not align in the absence of our alignment objective.

## F    ETHICS STATEMENT

Since our framework advances the capabilities of generative image synthesis, it shares the broader societal implications associated with deep generative models, such as the potential for misuse in creating misleading content, deepfakes, or copyright infringement. Furthermore, as our method aligns latent spaces to arbitrary pre-trained priors, there is a risk of propagating or amplifying biases inherent in those underlying foundation models.

