# OpenReview forum: "Aligning Latent Spaces with Flow Priors"
_ICLR.cc/2026/Conference — Submitted to ICLR 2026_

### Official Review · Reviewer_f6M9 · 2025-10-22

**Soundness:** 2
**Presentation:** 3
**Contribution:** 2
**Rating:** 4
**Confidence:** 4

**Summary:**

The works aims at aligning the latent space of a generative model to arbitrary prior distributions. The approach consist to (1) train a flow-matching model on features to have an estimate of the arbitrary prior (2) allign the latent space to this learnt prior through mean square error minimization of the latents.

The approach is evaluated through a toy example (mixture of 2D Gaussian) and image generation with a ViT based auto-encoder and ImageNet $256\times 256$ images, with four different types of priors modelled by a flow prior consisting of a 6 layer multi layer perceptron, the image generation being processed with a masked autoregressive model. The alignement is estimated by comparing the change during training of the proposed alignement loss with an estimate of the negative log likelihood (this last relying on k-NN density estimation). Then the results of the generation with these prior and a masked autoregressive model are evaluated according to several metrics reflecting the reconstruction and the quality of images, as well as their similarity to the ImageNet classes.

An ablation studies the robustness of the approach with regards to important hyper-parameters, using the prior with textual embeddings.

**Strengths:**

* the toy example with a mixture of five isotropic 2D Gaussian (section 5.1) nicely illustrated the proposed approach and helps the reader to understand its principle. The paper also demonstrates a genuine desire of clarity by providing an "intuitive explanation" of the method in section 4.2.

* for image generation, the experiments are conducted with four very different prior, namely low-level and semantic embedding (visual features), quantified visual features and even textual features.
  - the choice of a masked auto-regressive model is relevant, although one can regret that any experiment was conducted with diffusion-based models (but this choice is justified in the manuscript).
  - the experiment also independently estimates the incluence of classifier-free guidance in this context.

* the proposed approach allows theorically to get a tractable approach to maximize a variational lower bound on the log-likelihood of latents under the prior distribution. The formal proof is provided in a simple case in the mainpaper and further precise development are given in the appendix.

**Weaknesses:**

* results of alignment in section 5.2 (line 397-412) are made through the *observation* of two curves and commenting their (asuumed) correlation, without reporting this last. There is no comparison to any baseline.

* the result of image generation (section 5.2 and Table 1) are not really convincing. If one considers the results with classifier free guidance (that is, the best) the performance are close to the basic prior (AE, KL, SoftVQ) and sometimes worse, in particular in terms of Precicion and Recall. As well:
  - The results are reported for one set of generation only and it is thus to estiamte the significance of the results. It may have been relevant to estimate a standard deviation, at least for some of the models (e.g the AE and the Dino that gives the best results)
  - the fact that the textual feature lead to results comparable to other prior (best for IS withour CFG and Recall with CFG) is interpreted by authors (appendic D, line 71156-1162) as "[suggesting]  that high-quality textual representations capture abstract semantic structures that are transferable across modalities. The 896-D Qwen embeddings provide a rich semantic space that can effectively constrain the visual latent space without being overly restrictive.". However, this assertion is not convincing since, on the contrary, one could also consider that *any* prior could lead to similar results and that it mainly shows that the proposed approach does not model anything usefull. Hence it make arguable the initial assumption of the paper, namely that modeling an arbitrary prior is interesting
  - by the way, the results in terms od IS and FID for $256\times 256$ imageNet images is much better in the original paper of MAR: cf. Table 4 of (Li et al., 2024a) for AR/MAR-B: FID=3.48 and IS=192.4 without CFG and FID=2.31 IS=281.7 with CFG. The precision is also much better in both case. The proposed approach is only competitive in terms of recall whre it is better with CFG and on-par (depending on the prior...) without CFG.

* it is surprising to conduct the ablation studies (section 5.3) with the textual embedding (Qwen) as prior since it is not the model that performs best nor the most "natural" to choose to generate images (settings of Tab 1: cf. line 461)

**minor**:
  - on line 397-412, the value of $k$ (for $k$-nearest neighbors desity estimation) is not reported
  - several references are arxiv preprints while the article has been published e.g (Nichol and Dhariwal, 2021) at ICML 2021 or (Lipma et al., 2022) at ICLR 2022. All preprint that have been further published should refer to the reviewed paper.
  - the "simple" baselines AE, KL and SoftVQ (Table 1) are not presented in the paper
  - some assumptions in theoretical derivation are arguable
    - in the proof of proposition 1, it is assumed that $\lambda=1$ (line 282) that seems quite high in practice. For example, in the abblation study (Table 2) $\lambda$ is less than 0.05. However the appendix provides a proof for any arbitrary positive $\lambda$ for all steps.
    - in practice, assumption 1 (line 311) is also arguable since it assumes that the flow matchinging minimizes perfectly equation (3). This assumption seems important to have latents that goes to the prior. Thus, in practice, it would be interesting to estimate to xhich extent this assumption is valid.

**Questions:**

* what is the "standard toolkit" used to compute the FID and IS (line 1133)? More precisely, on which model does it rely? For example, (Esser et al. 2024) rely on CLIP L/14 but older works rely on Inception.
* what are the score in Table 2 if $\lambda=1$ as assumed in the proof of proposition 1 ?

**Details Of Ethics Concerns:**

Ethic concern is not addressed. Since the work can be applied to image synthesis in particular, there are many potential socieltal consequences of the work. One may, as (Esser et al, 2024) point to [a] for a discussion on these.

[a] Po, R., Yifan, W., Golyanik, V., Aberman, K., Barron, J.T., Bermano, A., Chan, E., Dekel, T., Holynski, A., Kanazawa, A., Liu, C.K., Liu, L., Mildenhall, B., Nießner, M., Ommer, B., Theobalt, C., Wonka, P. and Wetzstein, G. (2024), State of the Art on Diffusion Models for Visual Computing. Computer Graphics Forum, 43: e15063.

---

> ### Author Response · Authors · 2025-11-23
> **Response to Reviewer f6M9 - Part 1**
>
> We appreciate your constructive feedback and thorough review. We are glad that you found the **presentation good**, **the intuitive explanation clear**, and the **diverse choice of priors** relevant. Your comments helped us better clarify both the empirical and conceptual aspects of our work.
>
> **Q1. Results of alignment are made through the observation of two curves and commenting their (assumed) correlation, without reporting this last; there is no comparison to any baseline**
>
> Thank you for pointing out this limitation. We did not initially include a baseline because an unregularized Autoencoder (AE) inherently lacks the mechanism to align its latent space with arbitrary complex distributions (such as the diverse visual, textual, continuous, and discrete priors we explore). Without our alignment loss, the model has no information from the prior to guide it. To demonstrate this empirically, we have added a plot in Appendix D showing the k-NN distance metric $\\log r\_k(\\mathbf{z})$ for an AE with a Dino prior target but without the alignment loss. As expected, the alignment does not occur and actually degrades during training compared to our method.
>
> **Q2. The result of image generation (section 5.2 and Table 1) is not really convincing**
>
> We appreciate this concern. We focus on the results without Classifier-Free Guidance (CFG) as our primary metric because the CFG scale was optimized specifically for the KL-regularized model and kept constant for all other variants to ensure a fair comparison.
>
> It is important to clarify that while our framework successfully aligns the latent space to *arbitrary* prior distributions, this does not imply that *every* prior will outperform a standard Gaussian. The downstream generation quality depends on the properties of the prior selected. As discussed in Appendix D (L.1174), our results provide insight into which types of structured priors empirically yield better generation quality, validating the method's capability to facilitate such exploration.
>
> **Q3. The results are reported for one set of generation only and it is thus hard to estimate the significance of the results**
>
> We understand the importance of statistical significance and robustness. We did not originally report standard deviations due to the computational cost of the training pipeline on ImageNet. However, given the large scale of the dataset and the consistency across diverse priors, we believe the results are robust. To empirically validate this, we conducted three additional training runs for our best-performing setting, Semantic (Dino), using different random seeds. The results, reported below (w/o CFG), show very low variance, confirming the stability and significance of our findings.
>
> |                          | rFID      | PSNR       | FID        | IS          | Pre.      | Rec.      |
> | ------------------------ | --------- | ---------- | ---------- | ----------- | --------- | --------- |
> | Trial 1 (paper reported) | 1.26      | 23.07      | 11.47      | 101.74      | 0.59      | 0.59      |
> | Trial 2                  | 1.31      | 22.93      | 11.54      | 100.23      | 0.58      | 0.59      |
> | Trial 3                  | 1.23      | 23.15      | 11.20      | 105.82      | 0.60      | 0.58      |
> | Trial 4                  | 1.28      | 23.01      | 11.75      | 99.76       | 0.59      | 0.60      |
> | Overall                  | 1.27±0.03 | 23.04±0.08 | 11.49±0.20 | 101.89±2.38 | 0.59±0.01 | 0.59±0.01 |
>
> **Q4. The fact that the textual feature leads to results comparable to other priors may suggest that any prior could lead to similar results and mainly shows that the proposed approach does not model anything useful, making arguable the initial assumption of the paper that modeling an arbitrary prior is interesting**
>
> We appreciate this perspective, as it pushes us to clarify what is and is not implied by our results. As noted in our response to Q2, our method enables alignment to arbitrary priors, but it is not the case that any prior improves performance. As shown in Table 1, the discrete VQ features perform significantly worse than the other priors despite sharing some structural properties with the textual (Qwen) features.
>
> We maintain that the success of the Qwen embeddings is significant. It suggests that specific high-quality structured priors (like textual embeddings) can improve over the unregularized AE baseline, whereas others (like VQ) may not. This supports the claim that the method models useful structural information from the prior, rather than simply making the model indifferent to the choice of prior.

---

> ### Author Response · Authors · 2025-11-23
> **Response to Reviewer f6M9 - Part 2**
>
> **Q5. The results in terms of IS and FID for ImageNet images are much better in the original paper of MAR**
>
> Thank you for pointing out this comparison. The performance difference is primarily due to our choice of tokenizer configuration. The original MAR implementation uses an SD-VAE-f16, which produces 256 tokens of dimension 16 for a 256x256 image. In contrast, our tokenizer follows the SoftVQ design, producing only 64 tokens of dimension 32.
>
> We selected this configuration because we aimed for tokens that convey higher-level semantic information rather than low-level details. However, having fewer tokens with higher dimensionality makes the per-token prediction task significantly harder for the generative model compared to the original MAR setup. Our goal in this work was to validate the alignment framework across diverse priors, rather than to tune for the absolute best possible ImageNet scores under a specific architecture.
>
> **Q6. It is surprising to conduct the ablation studies (section 5.3) with the textual embedding (Qwen) as prior since it is not the model that performs best nor the most "natural" to choose to generate images**
>
> We understand this concern and appreciate the opportunity to clarify our choice. We selected the Qwen textual features for the ablation studies for two strategic reasons:
> 1.  Dimensionality: Qwen features have a much higher dimension (896) compared to the latent target (32). This large discrepancy is necessary to properly evaluate and ablate the effectiveness of the different downsampling methods.
> 2.  Training Efficiency: Unlike Dino features, which require extracting image features from the entire dataset during training, Qwen embeddings act as a fixed set of vectors, making the ablation training pipeline more efficient.
>
> This setting stressed the alignment component in a controlled way and allowed us to more clearly isolate the effect of design choices such as random projections.
>
> **Q7. What are the scores in Table 2 if $\\lambda = 1$ as assumed in the proof of proposition 1?**
>
> We thank you for highlighting this connection between the theory and practice. In our theoretical proof, we use $\\lambda=1$ to simplify the derivation. However, in practice, we set $\\lambda=0.01$ to balance the reconstruction objective against the regularization term, which is a standard practice in generative modeling (analogous to the $\\beta$ parameter in $\\beta$-VAE).
>
> If we set $\\lambda=1$ during training, the optimization is dominated by the alignment loss. The model prioritizes matching the prior distribution perfectly at the expense of the reconstruction loss, causing the latent space to collapse to the modes of the prior and failing to reconstruct the input image. As shown in Table 2(b), even increasing $\\lambda$ to 0.05 significantly degrades reconstruction. We experimentally confirmed that with $\\lambda=1$, reconstruction collapses completely.
>
> **Q8. Minor points and clarifications**
>
> *   Value of k: We used $k=5$ for the k-nearest neighbors density estimation. Thank you for noting this omission.
> *   Published references: We appreciate the detailed check and have updated all references to their published versions.
> *   Baselines not presented: Due to space constraints, we described the baselines primarily in the implementation details, as all models share the exact same architecture and differ only in the prior used. We have added a clearer explanation of the baselines in the main text.
> *   Toolkit for FID/IS: We utilize the evaluation toolkit from OpenAI's guided-diffusion. This toolkit relies on Inception-v3 features for metric calculation and matches the protocol used in the official MAR implementation.
>
> **Q9. Ethics concern is not addressed**
>
> Thank you for raising this important point. Indeed, as a generative image modeling framework, our method shares the common ethical considerations of the field. We have updated the paper to include a dedicated discussion on potential societal impacts in Appendix. F.

---

### Official Review · Reviewer_nxA8 · 2025-10-31

**Soundness:** 1
**Presentation:** 1
**Contribution:** 2
**Rating:** 2
**Confidence:** 4

**Summary:**

The paper proposes to use a pretrained flow model trained on the features of images to be the prior for a variational autoencoder.
The paper proposes to use a flow-matching-like objective using the pretrained flow model velocity field to approximate the prior log likelihood without running ODE solver to compute the likelihood.
The paper then runs experiments on synthetic dataset and ImageNet-1K using various fixed prior distributions.

**Strengths:**

- Simple (heuristic) objective that doesn't require ODE evaluation of flow models.
- Did an experiment on ImageNet-1K.

**Weaknesses:**

- "vθ encapsulates the dynamics that transport probability mass from the base distribution pinit to the prior distribution pprior along linear path" - I don't think the $v_\theta$ captures movement along linear paths. It is trained with linear paths but the velocity field itself does not produce a linear path. Only optimal transport maps would produce linear paths but that is not solved via flow matching.
  - Figure 2 while intuitive in this case will break down when the init and prior are overlapping but very different distributions. In this case, the flow matching vector field may be highly non-linear. I'm not convinced
  - Consider mixture of Guassians to mixture of Gaussians. A flow matching objective will map them to each other but the paths are not linear.
  - This seems to be an incorrect intuitive explanation for the objective.
  - "vθ precisely captures the dynamics required to transform initial noise samples x0 into prior features x along straight interpolation paths. Specifically, it has
learned to predict the exact velocity x − x0 at any point (1 − t)x0 + tx along such a path" - Again, I believe this is a misunderstanding of flow matching. Flow matching takes the average velocity over all possible pairings of points in a sense. It does not correctly predict the linear velocity since multiple pairs of points could produce the same latent.

- Proposition 1 seems vacuous since you don't compute $C(z)$ and (as far as I can tell) $C(z)$ could be positive or negative. Thus, $L_{align}$ does not form a lower bound even up to constants since $C(z)$ is not constant w.r.t. $z$. If it was constant w.r.t. $z$, then it would be okay for training the encoder. But as it stands, this bound isn't meaningful.
  - "and C(z) is dependent on z and vθ ." - This should be shown notationally as $C(z,\theta)$. This hides the fact that $C$ depends on $z$ and $\theta$.
  - "We analyze the behavior of C(z) in Appendix A to show that if z aligns with a more concentrated prior distribution (making Lalign(z; θ) small), C(z) tends to be positive and larger, contributing favorably to the ELBO." - Again, this highlights that $C(z)$ is not understood. Since $z$ depends on the encoder parameters, you cannot just ignore this $C(z)$ term.

- "Assumption 1 (Optimality of vθ ). The velocity field vθ : Rd1 × [0, 1] → Rd1 is (pre-trained) and optimal, satisfying vθ ((1 − t)x0 + tx1, t) = x1 − x0 ∀x0 ∼ pinit, x1 ∼ pprior, t ∈ [0, 1]." - Again as above, this is almost never  true for Flow matching velocity fields. Even if you use flow matching between two Gaussian distributions this will not be true. To make them "straight", you would have to do rectification steps like in rectified flow matching multiple times to converge on the optimal transport map. But, in general, flow matching objectives do not do this. This is a critically incorrect assumption.

- Fig 3--- This shows that the latents distribution does NOT converge to the prior distribution even for this toy example. But rather it converges towards the modes of the prior distribution. This aligns with the problems in the theory above.

- A broader motiviation question is why should the prior distributions be fixed? I'm still not convinced this is actually useful when learning an encoder. Can you provide a convincing example where you do an experiment and it directly improves some task? Like can you show that if you use Gaussian vs yours as a prior, it actually produces better robustness or classification accuracy for downstream tasks using linear probing? I'm not convinced arbitrary fixed priors are better than Gaussian fixed priors. It is more interesting like IAF to use learnable flexible priors but to have a fixed prior, it's not clear that this is actually helpful and is likely to hurt overall performance. This is important motivation issue with the proposed work.

**Questions:**

- How does this compare to the following relevant paper [Gong et la.,2025] on score-based priors for latent variable models (diffusion-based velocity fields but may be generalizable to flow-based velocity fields)? This paper also avoids running the ODE flow when training the encoder. If you just keep the velocity field fixed, then the encoder can be directly trained.

Gong, Z., Lim, J., & Inouye, D. I. Expressive Score-Based Priors for Distribution Matching with Geometry-Preserving Regularization. In Forty-second International Conference on Machine Learning.

---

> ### Author Response · Authors · 2025-11-23
> **Response to Reviewer nxA8 - Part 1**
>
> Thank you for your detailed critique and for acknowledging the **simplicity** of our objective and the **scale** of our ImageNet experiments. We greatly appreciate the care you put into analyzing the theoretical aspects, and we address your concerns and questions below.
>
> **Q1. "I don't think the $v_\\theta$ captures movement along linear paths. It is trained with linear paths but the velocity field itself does not produce a linear path. Figure 2 while intuitive in this case will break down when the init and prior are overlapping but very different distributions. This seems to be an incorrect intuitive explanation for the objective."**
>
> You are correct regarding the inference dynamics. The marginal vector field and resulting ODE trajectories are indeed curved. We would like to clarify the intended intuition based on the definition of the Flow Matching loss itself, rather than the resulting ODE trajectories.
>
> Our intuition follows directly from the Conditional Flow Matching (CFM) objective (Eq. 3 in the paper). In CFM, the model $v_\\theta$ is trained to regress the vector field $u_t(\\mathbf{x}|\\mathbf{x}_0, \\mathbf{x}_1) = \\mathbf{x}_1 - \\mathbf{x}_0$ along the conditional straight path $\\mathbf{x}_t = (1-t)\\mathbf{x}_0 + t\\mathbf{x}_1$. While the marginal vector field (and the resulting ODE trajectory) is indeed curved and non-linear as you correctly pointed out, the training target at any specific point on a conditional straight path is the straight velocity $\\mathbf{x}_1 - \\mathbf{x}_0$.
>
> Therefore, our alignment loss $L_{\\text{align}}$ effectively asks a reverse question: Given a latent $\\mathbf{z}$ and a noise $\\mathbf{x}_0$, is the vector $(\\mathbf{z} - \\mathbf{x}_0)$ consistent with what the pre-trained flow model $v\_\\theta$ expects at the interpolated points?
>
> *   If $\\mathbf{z}$ is a valid sample from $p_{\\text{prior}}$, the flow model (having seen similar training pairs) will predict a velocity close to $\\mathbf{z} - \\mathbf{x}_0$ along the path, resulting in low loss.
> *   If $\\mathbf{z}$ is an invalid sample ("bad" case in Fig. 2), the straight path connecting $\\mathbf{x}_0$ and $\\mathbf{z}$ traverses regions where the flow model predicts a different velocity field (pointing towards the true prior manifold), resulting in high loss.
>
> We acknowledge that our exposition could more clearly distinguish between the linear conditional paths used in the objective and the curved marginal paths of the resulting ODE. By "straight", we mean the straight path $\\mathbf{z} - \\mathbf{x}_0$ given a valid sample $\\mathbf{z}$, not that the ODE path predicted by the flow matching model is always straight. We believe that, in this sense, the mechanism of the loss—checking consistency against the straight interpolation paths used during training—provides a sound way to interpret the objective.

---

> ### Author Response · Authors · 2025-11-23
> **Response to Reviewer nxA8 - Part 2**
>
> **Q2. "Proposition 1 seems vacuous since you don't compute $C(\\mathbf{z})$ and $C(\\mathbf{z})$ could be positive or negative. Thus, the bound does not form a lower bound even up to constants since $C(\\mathbf{z})$ is not constant w.r.t. $\\mathbf{z}$. This hides the fact that $C(\\mathbf{z})$ depends on $\\mathbf{z}$ and $v\_\\theta$, and since $C(\\mathbf{z})$ depends on the encoder parameters, you cannot just ignore this $C$ term."**
>
> We appreciate this careful reading of our theoretical development. We explicitly acknowledge in Proposition 1 (L.263) and Eq. (10) that $C(\\mathbf{z})$ depends on both $\\mathbf{z}$ and $v\_\\theta$. While we do not compute $C(\\mathbf{z})$ during training due to its cost (it involves the trace of the Jacobian $\\nabla\_{\\mathbf{z}} v\_\\theta$), we view minimizing $\\mathcal{L}\_{\\text{align}}$ as a mathematically justified proxy for maximizing the ELBO, rather than treating it as a strict lower bound with a constant offset.
>
> *   Monotonicity: In Appendix A (Theorem 1), we formally prove that minimizing $\\mathcal{L}\_{\\text{align}}$ improves the ELBO under reasonable conditions. Specifically, if the reduction in $\\mathcal{L}\_{\\text{align}}$ is sufficiently large, it guarantees an increase in the log-likelihood lower bound, even accounting for changes in $C(\\mathbf{z})$.
> *   Behavior of $C(\\mathbf{z})$: The term $C(\\mathbf{z})$ represents the expected divergence of the flow field. As analyzed in Appendix A.2, this term behaves well for neural networks with bounded weights. Furthermore, our empirical results in Figure 3(c) show that $\\mathcal{L}\_{\\text{align}}$ and the true NLL are strongly correlated. As $\\mathcal{L}\_{\\text{align}}$ decreases, the true NLL decreases, confirming that the variation in $C(\\mathbf{z})$ does not negate the alignment signal.
> *   Tractable Proxy: The core contribution here is practical. Computing the exact ELBO requires expensive Jacobian trace calculations. We identify that the squared-error term ($\\mathcal{L}\_{\\text{align}}$) is the primary driver for alignment and can be optimized efficiently, serving as a tractable surrogate for the full likelihood objective.
>
> **Q3. "Fig 3 --- This shows that the latents distribution does NOT converge to the prior distribution even for this toy example. But rather it converges towards the modes of the prior distribution. This aligns with the problems in the theory above."**
>
> We appreciate this sharp observation, which is consistent with how we interpret the objective. Our alignment loss is derived as a proxy for maximizing the log-likelihood $\\log p\_{\\text{prior}}(\\mathbf{z})$ of latent samples.
>
> In the toy experiment (Figure 3), we minimize *only* the alignment loss without any input data or reconstruction constraints. In this unconstrained optimization setting, maximizing likelihood naturally drives samples toward the highest-density regions (modes) of the prior. The fact that the samples concentrate on the modes empirically supports that our loss is functioning as a likelihood maximizer.
>
> In the full autoencoder training (Section 5.2), this mode-seeking force is balanced by the reconstruction loss. The encoder must map diverse input images to distinct latent codes to preserve information. The alignment loss pulls latents onto the valid manifold, while the reconstruction loss spreads them across the manifold to cover the diverse inputs.
>
> This balance is exactly the motivation of our method: creating a latent space that is both structured and expressive. This is similar to VQ-VAE, where codebooks represent discrete modes, but our method provides a continuous regularization toward the prior manifold.

---

> ### Author Response · Authors · 2025-11-23
> **Response to Reviewer nxA8 - Part 3**
>
> **Q4. "A broader motiviation question is why should the prior distributions be fixed? I'm still not convinced this is actually useful when learning an encoder ... it's not clear that this is actually helpful and is likely to hurt overall performance. This is important motivation issue with the proposed work."**
>
> We appreciate this broader motivation question, which goes to the heart of why our framework may be useful. The key motivation is the **Reconstruction vs. Generation** dilemma.
>
> Gaussian priors have been working well, but they impose a trade-off: a strong prior (large KL weight) degrades reconstruction, while a weak prior (small KL weight) yields an unorganized latent space that is difficult for downstream generative models to learn. Modern approaches like SD-VAE often reduces the KL term to a very small term (1e-6) and Cosmos tokenizer removes the KL term completely, but this shifts the burden to the generative model.
>
> It is important to clarify that while our framework successfully aligns the latent space to *arbitrary* prior distributions, this does not imply that *arbitrary* prior will be better than a standard Gaussian. Our method allows us to use semantic, pre-trained priors that are inherently more structured than a Gaussian. To demonstrate the utility of this alignment on downstream tasks beyond generation, we conducted a linear probing experiment based on your suggestion. We trained a linear classification head on the frozen latent space of our trained Autoencoders using ImageNet-1k for 10 epochs:
>
> |Autoencoder|Top-1 Acc|
> |:--|:--|
> |AE (No prior)|29.4|
> |KL (Gaussian)|34.5|
> |Dino Aligned|42.1|
>
> These results indicate that aligning with Dino features improves the linear separability of the latent space compared to a standard Gaussian prior. This supports that our method can transfer the structural benefits of the semantic prior to the learnable latent space, benefiting downstream discriminative tasks as well as generative ones.
>
> Finally, we wish to highlight that utilizing fixed, high-quality feature spaces to guide latent learning is now a common and successful paradigm in recent work. Methods such as VA-VAE, SoftVQ-VAE, and RobustTok all align latent tokens with powerful pre-trained semantic encoders and consistently report improved robustness and generative quality over purely reconstruction-driven VAEs [1–3]. More recently, RAE show that directly replacing standard VAE encoders with frozen, pre-trained representation backbones is very effective for diffusion transformers, further underscoring the trend toward fixed, semantically structured priors [4].
>
> [1] J. Yao and X. Wang, “Reconstruction vs. Generation: Taming Optimization Dilemma in Latent Diffusion Models,” CVPR 2025.
> [2] H. Chen et al., “SoftVQ-VAE: Efficient 1-Dimensional Continuous Tokenizer,” arXiv:2412.10958, 2024.
> [3] K. Qiu et al., “Robust Latent Matters: Boosting Image Generation with Sampling Error Synthesis,” arXiv:2503.08354, 2025.
> [4] B. Zheng, N. Ma, S. Tong, and S. Xie, “Diffusion Transformers with Representation Autoencoders,” arXiv:2510.11690, 2025.
>
> **Q5. "How does this compare to the following relevant paper [Gong et la.,2025] on score-based priors for latent variable models (diffusion-based velocity fields but may be generalizable to flow-based velocity fields)? This paper also avoids running the ODE flow when training the encoder. If you just keep the velocity field fixed, then the encoder can be directly trained."**
>
> We thank you for pointing out this closely related work. SFS shares a similar motivation with our work: to avoid intractable computations like solving ODEs or calculating Jacobians at each step, but there are several important differences:
>
> *   Stability: SFS (Gong et al.) relies on score matching. As discussed in their own Appendix C, SFS faces instability at low noise levels (where scores explode), requiring specific stabilization techniques. In contrast, our method relies on Flow Matching. The target velocity in optimal transport paths is $\\mathbf{x}\_1 - \\mathbf{x}\_0$, which is bounded and well-behaved at all time steps $t \\in [0,1]$. Consequently, our alignment loss does not suffer from exploding gradients and does not require hyperparameter tuning for noise levels.
> *   Methodology: SFS substitutes a learned score function into the alignment objective. Our method reformulates the flow matching objective itself.
> *   Inference: SFS optimizes by following the gradient of the log-density (score). Our method optimizes by minimizing the drift error, ensuring the latent $\\mathbf{z}$ is a consistent endpoint of a probability flow trajectory starting from noise.

---

### Official Review · Reviewer_AhAf · 2025-11-01

**Soundness:** 3
**Presentation:** 3
**Contribution:** 4
**Rating:** 6
**Confidence:** 2

**Summary:**

The paper proposes a novel method for aligning a learnable latent space with any arbitrary prior distribution. It first trains a flow matching model to model the prior distribution. It then optimizes the target latents to minimize an alignment loss computed by the flow model. The paper provides theoretical and empirical analyses to support the proposed technique. In particular, in the experiments, it successfully aligns latents with a ViT-L-based encoder-decoder to 4 different prior distributions, including low-level visual features from a VAE, continuous semantic visual features from DinoV2, discrete visual codebook embeddings from LlamaGen VQ, and textual embeddings from Qwen. These latents can be integrated into MAR-B models, yielding competitive conditional generation results on ImageNet 256x256.

**Strengths:**

- The paper handles an interesting problem using a novel mechanism
- The paper provides theoretical and empirical analyses to support the proposed technique
- The proposed method has high practical value and can be applied in a wide range of problems. In the experiments, the paper demonstrates it by aligning simple latents formed by ViT-L-based encoder-decoder systems to 4 different prior distributions, including low-level visual features from a VAE, continuous semantic visual features from DinoV2, discrete visual codebook embeddings from LlamaGen VQ, and textual embeddings from Qwen.
- The aligned latents can be integrated into MAR-B models, yielding competitive conditional generation results on ImageNet 256x256.
- To solve the dimension mismatching issue, it shows that among linear projection options, Random Projection is surprisingly a strong and more stable option

**Weaknesses:**

- The method requires two training steps to align the space; thus, it aggregates errors from both steps. First, it depends heavily on the quality of the flow-matching model, which cannot capture the prior distribution perfectly. Second, the latent optimization process is also not guaranteed to converge. The authors should analyze error accumulation and the system's failure modes.
- The method requires two training steps, which are expensive. Computation cost should be reported.
- While the proposed method can align the latent spaces to the target space in terms of distribution, it is interesting to incorporate it with semantic alignment. For example, when aligning the ViT-L-encoder-based latent space to the textual embeddings from Qwen, it is better to ensure that the encoded textual embedding aligns with the input image content.

**Questions:**

- The method requires two training steps to align the space; thus, it aggregates errors from both steps. First, it depends heavily on the quality of the flow-matching model, which cannot capture the prior distribution perfectly. Second, the latent optimization process is also not guaranteed to converge. The authors should analyze error accumulation and the system's failure modes.
- The method requires two training steps, which are expensive. Computation cost should be reported.
- While the proposed method can align the latent spaces to the target space in terms of distribution, it is interesting to incorporate it with semantic alignment. For example, when aligning the ViT-L-encoder-based latent space to the textual embeddings from Qwen, it is better to ensure that the encoded textual embedding aligns with the input image content.

---

> ### Author Response · Authors · 2025-11-23
> **Response to Reviewer AhAf**
>
> Thank you for your time and valuable feedback! We are grateful for your positive assessment of our work. We particularly appreciate your recognition of the **novelty of our framework**, the **theoretical and empirical rigor**, and the **high practical value**. We are also glad you found the random projection analysis insightful. We address your concerns and questions below:
>
> **Q1. The method requires two training steps to align the space; thus, it aggregates errors from both steps. First, it depends heavily on the quality of the flow-matching model, which cannot capture the prior distribution perfectly. Second, the latent optimization process is also not guaranteed to converge. The authors should analyze error accumulation and the system's failure modes.**
>
> We agree that a two-stage process introduces potential for error accumulation, and we appreciate you highlighting this concern. However, our method is designed to be highly robust to these risks in practice:
>
> - Robustness of Flow Matching in Stage 1: The first stage involves training a flow matching model to map a standard Gaussian source to the ground-truth feature distribution (e.g., DinoV2). This is a standard generative modeling task and is known to be very stable. We trained the model for 1 million steps to ensure the model fully converged. Even if the flow model is not perfect, it effectively captures the underlying manifold structure required for regularization.
> - Stationary Optimization Landscape: In the second stage, the flow model is frozen. Our flow model acts as a fixed, deterministic "critic." This stationarity makes the optimization landscape for the autoencoder significantly more stable compared to adversarial training where the discriminator changes dynamically.
> - Role as Regularization: The alignment loss serves only as a regularizing term. In the second stage, the tokenizer is jointly trained with reconstruction loss, adversarial loss, and perceptual loss, which guarantee its convergence. We do not require perfect alignment; the flow model provides structural guidance as long as it is well-trained.
>
> Regarding failure modes, we explicitly analyzed this in Section 5.2. The only observed divergence case in our experiments is the VQ (Discrete) case. This failure occurred because the low dimensionality (8-dim) of the discrete LlamaGen code lacked the capacity to satisfy the alignment constraints, causing the loss to collapse. In all other experiments, the alignment loss consistently decreases alongside the reconstruction loss.
>
> **Q2. The method requires two training steps, which are expensive. Computation cost should be reported.**
>
> We agree that efficiency is crucial and is actually an advantage of our method. Our method is computationally lightweight:
>
> - Stage 1 One-time Cost: Training the MLP flow prior takes only a few hours on a single node (8 GPUs). This is negligible compared to the resources required for training the autoencoder or the subsequent generative model, which takes about 5 days in total.
> - Stage 2 Training Cost: During autoencoder training, our alignment loss requires only a single forward pass through the MLP flow model per iteration. We do not solve ODEs backwards through time. The additional computation has almost zero impact on the training speed.
>
> We have added a breakdown of these costs in Appendix C to provide a clearer comparison. Thank you for pointing this out.
>
> **Q3. While the proposed method can align the latent spaces to the target space in terms of distribution, it is interesting to incorporate it with semantic alignment. For example, when aligning the ViT-L-encoder-based latent space to the textual embeddings from Qwen, it is better to ensure that the encoded textual embedding aligns with the input image content.**
>
> This is an excellent insight, and we appreciate the suggestion. While semantic alignment is indeed a promising direction, we believe it is beyond the scope of our current method for the following reasons:
>
> - Motivation: Our primary goal is per-token regularization to improve the generative quality of the latent space. Semantic corrspondence would require per-sample regularization.
> - Data Constraints: Strict semantic alignment would typically require image-text paired data to align specific image tokens with specific text embeddings, which is not available in the ImageNet setting we studied.
>
> Nevertheless, we agree that semantic alignment is a valuable direction for future work, similar to explorations in recent methods like TA-TiTok [1].
>
> [1] Kim, Dongwon, et al. "Democratizing text-to-image masked generative models with compact text-aware one-dimensional tokens." ICCV 2025.

---

> > ### Comment · Reviewer_AhAf · 2025-11-28
> >
> > Thank you for the rebuttal. It addressed my concerns pre-rebuttal.

---

### Author Response · Authors · 2025-12-03
**Summary of Reviews and Rebuttals**

We thank the reviewers and ACs for their valuable time in evaluating our work. Below we summarize each review and how our rebuttal addressed the main concerns.

### Reviewer AhAf (rating: 6, conf: 2)

Reviewer AhAf acknowledged the novelty of our framework, the theoretical and empirical support, its high practical value across diverse priors, and the usefulness of our analysis on dimension mismatch, especially the strength of random projection. After rebuttal, the reviewer stated that their concerns were addressed.

**Two-stage error accumulation**: We clarified that the flow prior is trained to convergence as a standard, stable generative model; in the second stage it is frozen and used only as a deterministic regularizer alongside reconstruction, adversarial, and perceptual losses. We also explicitly reported observed failure modes (only in the VQ case).

**Two-stage training cost**: We reported exact compute: the flow prior is a one-time, few-hour MLP training on a single node (8 GPUs), and the alignment loss in stage two is just a single forward pass, adding negligible overhead.

**Semantic alignment with text priors**: We explained that our focus is per-token distributional regularization, whereas semantic alignment requires per-sample supervision and paired image–text data, which ImageNet does not provide. We positioned semantic alignment as an important but out-of-scope direction for future work.

### Reviewer nxA8 (rating: 2, conf: 4)

Reviewer nxA8 appreciated the simplicity of the objective (no ODE solving) and the ImageNet-1K experiments, but raised strong concerns about the theoretical formulation and the motivation. We believe these concerns stem from a misunderstanding of the alignment objective; unfortunately, we did not receive a response during the discussion.

**“Straight paths”**: We clarified the "straight path" is based on the definition of flow matching loss, rather than the resulting ODE trajectories. Our intuition is based on the flow matching objective: the model is trained to regress the straight-line conditional velocity x₁ − x₀ along linear interpolation paths, while the actual marginal ODE trajectories are indeed curved.

**Role of C(z) and the ELBO proxy**: We emphasized that C(z) is explicitly acknowledged as depending on z and v_θ and is not treated as a constant. We position L_align as a tractable surrogate for the full ELBO, not a strict lower bound. In Appendix A we prove that, under reasonable conditions, reducing L_align improves the ELBO, and we show empirically that L_align is strongly correlated with the true NLL.

**Mode-seeking behavior in the toy example**: We explained that this is expected in the toy setting and aligned with our theory. In the full autoencoder, this mode-seeking is balanced by reconstruction.

**Motivation for fixed non-Gaussian priors**: We framed our method as a way to mitigate the reconstruction–generation trade-off by aligning to semantic priors. To make this concrete, we added a linear probing experiment on ImageNet, showing improved downstream performance. We also connected our approach to recent work (e.g., VA-VAE, SoftVQ-VAE, RobustTok, RAE) that similarly leverages fixed, semantically structured spaces.

### Reviewer f6M9 (rating: 4, conf: 4)

Reviewer f6M9 appreciated the clarity of the intuitive explanation, the illustrative toy example, the use of four very different priors, the relevance of using a MAR-B backbone, and the theoretical connection to a tractable variational objective. The main concerns centered on the strength of the empirical evidence, the comparison to MAR, and whether arbitrary priors are truly useful. We addressed these points in detail but did not receive a follow-up during the discussion.

**Lack of alignment baselines and correlation quantification**: We added an AE baseline that uses the same target prior (e.g., Dino) but no alignment loss, and showed that the k-NN density metric does not improve and can even worsen compared to our method.

**Strength and stability of generation results**: We explained that our primary comparison is without classifier-free guidance, since the CFG scale was tuned for the KL model and reused for all variants. To address robustness, we ran three additional seeds for the best setting and reported low variance across metrics.

**Comparison to original MAR**: We clarified that our weaker FID/IS vs. MAR mainly comes from a different tokenizer: we use a SoftVQ-style tokenizer with 64×32 tokens instead of SD-VAE f16 with 256×16 tokens, making the task harder but encouraging semantic tokens. Our goal here was to study alignment across diverse priors, not to optimize MAR’s absolute scores.

**Interpretation of textual priors and broader usefulness of arbitrary priors**: We argued that comparable performance with Qwen text features does not mean “any” prior works: discrete VQ priors perform significantly worse, while Dino and Qwen improve over the unregularized AE baseline.

---

### Meta-Review · Area_Chair_57Yy · 2026-01-07

**Summary:**

This paper receives 1x marginal accept, 2x marginal rejects and 1x reject. The reviews pointed out major flaws in the formulation of the proposed method. There are also flawss in the assumption which can never be true. The experiments results are not convincing and some of them are showing otherwise to what is claimed in the paper. The AC agrees with the reviewers to reject the paper.

**Reviewer Concerns:**

The reviews pointed out major flaws in the formulation of the proposed method. There are also flawss in the assumption which can never be true. The experiments results are not convincing and some of them are showing otherwise to what is claimed in the paper. All the weaknesses are still outstanding after rebuttal.

**Reviewer Scores:**

The reviewers would maintain their scores.

---

### Decision · Program_Chairs · 2026-01-26

Reject